# ADePT: Adaptive Decomposed Prompt Tuning for Parameter-Efficient Fine-tuning

**Pengwei Tang, Xiaolin Hu, Yong Liu**[*]
Renmin University of China, Beijing, China
`{tangpwei,xiaolinhu,liuyonggsai}@ruc.edu.cn`

## ABSTRACT

Prompt Tuning (PT) enables the adaptation of Pre-trained Large Language Models (PLMs) to downstream tasks by optimizing a small amount of soft virtual tokens, which are prepended to the input token embeddings. Recently, Decomposed Prompt Tuning (DePT) has demonstrated superior adaptation capabilities by decomposing the soft prompt into a shorter soft prompt and a pair of low-rank matrices. The product of the pair of low-rank matrices is added to the input token embeddings to offset them. Additionally, DePT achieves faster inference compared to PT due to the shorter soft prompt. However, in this paper, we find that the position-based token embedding offsets of DePT restrict its ability to generalize across diverse model inputs, and that the shared embedding offsets across many token embeddings result in sub-optimization. To tackle these issues, we introduce **A**daptive **De**composed **P**rompt **T**uning (ADePT), which is composed of a short soft prompt and a shallow token-shared feed-forward neural network. ADePT utilizes the token-shared feed-forward neural network to learn the embedding offsets for each token, enabling adaptive embedding offsets that vary according to the model input and better optimization of token embedding offsets. This enables ADePT to achieve superior adaptation performance without requiring more inference time or additional trainable parameters compared to vanilla PT and its variants. In comprehensive experiments across 23 natural language processing tasks and 4 typical PLMs of different scales, ADePT consistently surpasses the other leading parameter-efficient fine-tuning methods, and even outperforms the full fine-tuning in certain scenarios. We also provide a theoretical analysis towards ADePT. Code is available at `https://github.com/HungerPWAY/ADePT`.

## 1 INTRODUCTION

Recently, Pre-trained Large Language Models (PLMs) (Raffel et al., 2020; Touvron et al., 2023) have seen rapid development, with commonly used models now on the scale of hundreds of millions and billions of parameters. Full fine-tuning (FT) of these PLMs requires substantial GPU resources, which is a common challenge faced by both academia and industry. To alleviate this resource-intensive issue, Parameter-Efficient Fine-Tuning (PEFT) (Houlsby et al., 2019; Liu et al., 2022; Hu et al., 2021; Ben Zaken et al., 2022) methods have gained significant attention and have seen breakthrough progress. These PEFT methods tune only a small amount of the internal parameters of a model or extra parameters, allowing PLMs to adapt effectively to target downstream tasks while maintaining performance comparable to FT.

The vanilla Prompt Tuning (PT) (Lester et al., 2021) uses a trainable soft prompt prepended to the input token embeddings (Lester et al., 2021), as shown in Figure 1a. The few trainable parameters make PT one of the mainstream methods for parameter-efficient fine-tuning. The improvements to PT can be categorized into four paths: the first path involves adding soft prompts to each layer of one PLM (Li & Liang, 2021); the second path involves stabilizing the optimization of soft prompt through a shallow network with a residual connection (Razdaibiedina et al., 2023); the third path involves using soft prompts that had already been trained by other methods for transfer learning (Vu et al., 2022; Asai et al., 2022; Wang et al., 2023); the fourth path uses input token embedding offsets

---

[*]Corresponding Author.

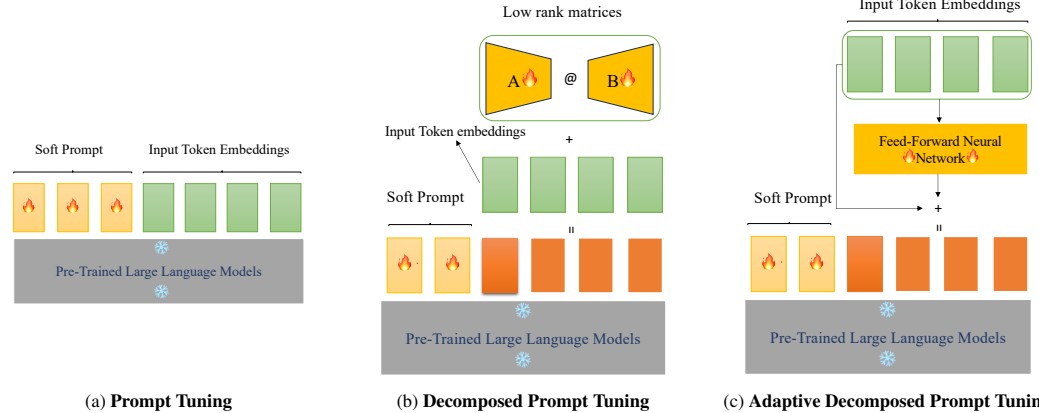

Figure 1: The overview of Prompt Tuning (PT), Decomposed Prompt Tuning, and Adaptive Decomposed Prompt Tuning (ADePT). PT uses a soft prompt prepended to input token embeddings. DePT uses a short soft prompt and offsets the input token embeddings using a pair of low-rank matrices. ADePT uses a short soft prompt and offsets the input token embedding using a token-shared shallow feed-forward neural network. ADePT can adaptively give input token embedding offsets based on input tokens, while DePT can only give position-based input token embedding offsets. Moreover, the use of a short soft prompt makes DePT and ADePT faster during inference.

to map the input token embedding into better embedding space (Shi & Lipani, 2024). The first three approaches either require increasing trainable parameters or necessitate additional transfer learning, while the fourth approach requires neither. For the fourth path, Decomposed Prompt Tuning (DePT) (Shi & Lipani, 2024) pioneers the shift in attention from soft prompt to applying input token embeddings offsets to the input token embeddings. As shown in Figure 1b, DePT learns embedding offsets $\Delta \boldsymbol{E} = \boldsymbol{AB} = [\Delta \boldsymbol{e}_1, \Delta \boldsymbol{e}_2, \cdots, \Delta \boldsymbol{e}_s]$ by optimizing a pair of low-rank matrices $\boldsymbol{A} \in \mathbb{R}^{s \times r_s}$ and $\boldsymbol{B} \in \mathbb{R}^{r_s \times d}$, where $s$ denotes the length of input tokens, $d$ denotes the dimension of input tokens and $r_s$ denotes the maximum rank of matrices $\boldsymbol{A}$ and $\boldsymbol{B}$. Given the input token embeddings $\boldsymbol{E} = [\boldsymbol{e}_1, \boldsymbol{e}_2, \cdots, \boldsymbol{e}_s] \in \mathbb{R}^{s \times d}$, the updated token embeddings $\boldsymbol{E}'$ are:

$$\boldsymbol{E}' = \boldsymbol{E} + \Delta \boldsymbol{E} = \boldsymbol{E} + \boldsymbol{AB}$$

DePT also learns a short soft prompt $\boldsymbol{P}_s \in \mathbb{R}^{l_s \times d}$ with length $l_s$. The input token embeddings $\boldsymbol{E}$, after DePT processing, are formulated as $[\boldsymbol{P}_s, \boldsymbol{E} + \boldsymbol{AB}]$. In this paper, we focus on improving the fourth path by providing better offsets for the input token embeddings. Additionally, our improvement of the fourth path is orthogonal to the first three and can be integrated with their methods.

The main contribution of DePT is the use of learnable token embedding offsets, which map the input token embeddings into a more suitable embedding space. DePT has demonstrated promising results across various tasks and PLMs, but it still has limitations. The updated token embeddings are formulated as $\boldsymbol{E} + \Delta \boldsymbol{E} = [\boldsymbol{e}_1 + \Delta \boldsymbol{e}_1, \boldsymbol{e}_2 + \Delta \boldsymbol{e}_2, \cdots, \boldsymbol{e}_s + \Delta \boldsymbol{e}_s]$. We can observe that in a target downstream task, the input token embeddings $\boldsymbol{E}$ vary, while the offsets $\Delta \boldsymbol{E}$ are position-based and fixed after fine-tuning. For a token $\boldsymbol{e}$, its position in the task can be arbitrary, meaning that both $\boldsymbol{e} + \Delta \boldsymbol{e}_i$ and $\boldsymbol{e} + \Delta \boldsymbol{e}_j$ can be its updated embedding in this task, where $1 \leq i \leq s, 1 \leq j \leq s, i \neq j$. This means that within the same task, the same token may have multiple different token embeddings, implying that the token embeddings of DePT violate the uniqueness of token embeddings. In Section 3.2, we design two experiments that reveal two limitations of DePT: (1) the offsets obtained by tokens at different positions may lead to different prediction outcomes; (2) to ensure that token embeddings are as unique as possible, the offsets in DePT are often much smaller than the input token embeddings, leading to sub-optimization.

Based on the above analysis, the embedding offset for each token should be the same at any position. Additionally, to enhance expressiveness, the offset corresponding to each token should be as unique as possible. Surprisingly, we find that using a function of the token embedding can meet the above requirements, *i.e.*, using the function $f(\boldsymbol{e})$ to be the offsets of input token embedding $\boldsymbol{e}$. To avoid increasing the number of trainable parameters compared to vanilla PT, we utilize a shallow and narrow feed-forward neural network to approximate the function for offset prediction. As shown in Figure 1c, we propose **A**daptive **De**composed **P**rompt **T**uning (ADePT), which uses a short soft

prompt and a two-layer feed-forward neural network, where the short soft prompt is prepended to the input token embeddings and the feed-forward neural network is shared by all tokens and produces the offset of each token. The use of a short prompt makes ADePT achieve faster inference speeds than the vanilla PT and comparable inference speeds compared to DePT.

In summary, the main contributions of this paper are as follows:

- We point out that the limitations of position-based token embedding offsets in DePT.To tackle these issues, we propose Adaptive Decomposed Prompt Tuning (ADePT), which can produce unique token embedding offset for each token.
- ADePT employs token-shared feed-forward neural networks to learn a unique offset for each token, which can adaptively adjust the token embedding offsets based on model input and allow the training parameters to achieve better optimization.
- Extensive evaluations across 23 NLP tasks and 4 PLMs of different scales show that ADePT surpasses leading PEFT methods, including full fine-tuning in certain scenarios. We theoretically analyze that ADePT possesses better expressive power than PT and DePT.

## 2   RELATED WORKS

**Parameter-Efficient Fine-tuning:** PEFT methods adapt PLMs to downstream tasks by optimizing a small number of parameters, significantly reducing computational resource demands. PEFT methods can be divided into four types:(1) Adapter and its variants (Houlsby et al., 2019; He et al., 2022; Rücklé et al., 2021; Ivison & Peters, 2022); (2) Low-rank Adaptation (LoRA) and its variants (Hu et al., 2021; Liu et al., 2022; Kopiczko et al., 2024); (3) Prompt Tuning (PT) and its variants (Lester et al., 2021; Li & Liang, 2021; Vu et al., 2022; Asai et al., 2022; Wang et al., 2023; Ma et al., 2022; Xiao et al., 2023; Razdaibiedina et al., 2023; Shi & Lipani, 2024); (4) other methods (Ben Zaken et al., 2022; Guo et al., 2021). The adapter inserts new modules into the transformer blocks (Houlsby et al., 2019). Hyperformer (He et al., 2022) uses a shared hypernetwork to generate task-conditioned adapters, reducing the trainable parameters in multi-task scenarios. AdapterDrop (Rücklé et al., 2021) removes adapters from lower transformer layers during training and inference, which reduces the computation overhead. Hyperdecoders (Ivison & Peters, 2022) generate input-specific adapters using a shared decoder for multi-task scenarios. LoRA (Houlsby et al., 2019) adopts the matrix product of a pair of low-rank matrices to approximate the updates of corresponding parameters. $(IA)^3$ (Liu et al., 2022) rescales internal activations only using learned vectors injected into the attention and feedforward modules. Prompt tuning (PT) (Lester et al., 2021)adapts PLMs to the new downstream tasks by optimizing learnable virtual tokens. LST (Sung et al., 2022) reduces memory requirements by training a small side network, achieving adaptation without back-propagating through the main network. BitFit (Ben Zaken et al., 2022) only tunes the bias of PLMs, which extremely reduces the trainable parameters. Diff pruning learns a sparse "diff" vector to modify a small percentage of pre-trained parameters, enabling efficient adaptation (Guo et al., 2021). Our proposed ADePT is an improved variant of PT, which can also be unified with other PEFT methods.

**Prompt tuning and its variants:** PT (Lester et al., 2021) enables the adaptation of PLMs to downstream tasks by learning a soft prompt appended in front of model input. Prefix-tuning (Li & Liang, 2021) can be viewed as an extension of Prompt Tuning applied across the entire depth of the model. SPoT (Vu et al., 2022) utilizes trained prompts from source tasks as initialization, but it requires extensive search to find the optimal initialization. ATTEMPT (Asai et al., 2022) adopts an attention module to compose the trained prompts of source tasks. MPT (Wang et al., 2023) learns a transferable prompt by distilling from multiple task-specific prompts. XPrompt (Ma et al., 2022) empirically demonstrates the negative impact of trained prompt tokens and proposes a hierarchical structured pruning for a trained soft prompt, which can be seen as a post-training method. DPT (Xiao et al., 2023) adopts the product of two low-rank matrices to approximate the soft prompt. Residual Prompt Tuning (Razdaibiedina et al., 2023) reparameterizes soft prompt embeddings with a shallow feed-forward network and residual connection, while ADePT uses the feed-forward network to learn input token embedding offsets—a distinct objective. DePT (Shi & Lipani, 2024) first discovers that the input token embedding offsets can enhance the performance of PT and uses short soft prompts to accelerate inference. ADePT produces adaptive input token embedding offsets via shallow token-shared networks, addressing the limitations of DePT mentioned in Section 1. With a comparable number of trainable parameters, ADePT can outperform both vanilla PT and DePT.

Table 1: The comparisons between DePT and ADePT on RTE and BoolQ tasks. All results are based on the T5-base model. The numbers within "DePT()" indicate the amount of cyclic left shift applied to the position-based input embedding offsets of DePT.

|        | DePT (0) | DePT (50) | DePT(100) | DePT(150) | DePT(200) | ADePT |
|--------|----------|-----------|-----------|-----------|-----------|-------|
| RTE    | 79.1     | 78.4      | 79.9      | 79.1      | 78.4      | 82.0  |
| BoolQ  | 78.4     | 74.7      | 73.1      | 70.9      | 73.9      | 80.2  |

Table 2: The Mean and Variance of input token embeddings and embedding offsets of DePT and ADePT on RTE and BoolQ tasks. All results are based on the T5-base model. The Mean and Variance are calculated from the tokens of the entire training dataset. The "mean()" and "variance()" refer to the corresponding task inside the parentheses.

|                                        | Mean (RTE) | Variance (RTE) | Mean (Boolq) | Variance (Boolq) |
|----------------------------------------|------------|----------------|--------------|------------------|
| Input Embeddings                       | 6.07       | 16.29          | 7.93         | 9.98             |
| The offset (absolute value) of DePT    | 0.01       | 0.06           | 0.02         | 0.01             |
| The offset (absolute value) of ADePT   | 8.31       | 5.45           | 6.09         | 3.70             |

## 3 METHOD

In this section, we revisit the preliminaries of PT and DePT (Section 3.1) and analyze the limitations of DePT (Section 3.2). Then, we introduce our proposed Adaptive Decomposed Prompt Tuning (ADePT) in Section 3.3, followed by a theoretical analysis of ADePT in Section 3.4.

### 3.1 PRELIMINARIES: PROMPT TUNING (PT) AND DECOMPOSED PROMPT TUNING (DEPT)

**Prompt Tuning(PT).** Let $\mathcal{D} = \{\boldsymbol{X_i}, \boldsymbol{y_i}\}_{i=1}^{N}$ be the training dataset of the target downstream task $\mathcal{T}$, where $N$ is the number of the training data. Given a PLMs with parameters $\Theta$, each input $\boldsymbol{X}_i$ is first mapped to token embeddings $\boldsymbol{E}_i \in \mathbb{R}^{s \times d}$ by tokenizer and embedding layer, where $s$ denotes the maximum length of input tokens and $d$ denotes the dimension of the input token embeddings. The objective of PT is to learn a soft prompt $\boldsymbol{P} \in \mathbb{R}^{l \times d}$ to enable the adaptation of a PLM to the target downstream task. Here, $l$ denotes the length of the soft prompt. The soft prompt $\boldsymbol{P}$ is prepended to the input token embeddings $\boldsymbol{E}$. The loss of PT for the target downstream task is formulated as:

$$\mathcal{L}_{\text{PT}} = -\sum_{i} \log P(\boldsymbol{y}_i \mid [\boldsymbol{P}, \boldsymbol{E}_i]; \Theta), \tag{1}$$

where $\mathcal{L}_{\text{PT}}$ is the loss function only optimized with regard to the soft prompt $\boldsymbol{P}$.

**Decomposed Prompt Tuning (DePT).** DePT adapts PLMs to the new target downstream task via a short soft prompt $\boldsymbol{P}_{s_1} \in \mathbb{R}^{l_{s_1} \times d}$ and a pair of low-rank matrices, *i.e.*, $\boldsymbol{A} \in R^{s \times r_s}$ and $\boldsymbol{B} \in \mathbb{R}^{r_s \times d}$, where $l_{s_1}$ denotes the length of short soft prompt and $r_s \ll \min(s, d)$ denotes the maximum rank of matrices $\boldsymbol{A}$ and $\boldsymbol{B}$. Similar to PT, the short soft prompt of DePT is prepended to the frozen input token embeddings. The product of low-rank matrices of DePT is used as the offsets of the input word embeddings. The loss of DePT for the target downstream task is formulated as:

$$\mathcal{L}_{\text{DePT}} = -\sum_{i} \log P(\boldsymbol{y}_i \mid [\boldsymbol{P}_{s_1}, \boldsymbol{E}_i + \boldsymbol{A}\boldsymbol{B}]; \Theta), \tag{2}$$

where the loss function $\mathcal{L}_{\text{DePT}}$ is optimized only with respect to the short soft prompt $\boldsymbol{P}_{s_1}$ and the pair of low-rank matrices $\boldsymbol{A}$ and $\boldsymbol{B}$.

### 3.2 THE LIMITATIONS OF DEPT

In this section, we explore several key factors that limit the performance of DePT.

Let $\boldsymbol{A}\boldsymbol{B} = \Delta\boldsymbol{E}$, the updated input token embeddings of DePT is formulated as $[\boldsymbol{e}_1 + \Delta\boldsymbol{e}_1, \boldsymbol{e}_2 + \Delta\boldsymbol{e}_2, \cdots, \boldsymbol{e}_s + \Delta\boldsymbol{e}_s]$. For a downstream task, the input token embeddings $\boldsymbol{E}$ vary. However, the

pair of low-rank matrices are fixed after adaptation, meaning that the input token embedding offsets $\Delta \boldsymbol{E}$ are position-based. Assume there are input token embeddings $[\boldsymbol{a}, \boldsymbol{b}, \boldsymbol{c}]$ (here, we omit the right padding), the updated token embeddings are formulated as $[\boldsymbol{a} + \Delta \boldsymbol{e}_1, \boldsymbol{b} + \Delta \boldsymbol{e}_2, \boldsymbol{c} + \Delta \boldsymbol{e}_3]$. Assume we have a meaningless sequence that does not affect the prediction performance, denoted as $[\boldsymbol{t}_1, \boldsymbol{t}_2]$. Prepending the meaningless sequence to input token embeddings $[\boldsymbol{a}, \boldsymbol{b}, \boldsymbol{c}]$, the updated input token embeddings are $[\boldsymbol{t}_1 + \Delta \boldsymbol{e}_1, \boldsymbol{t}_2 + \Delta \boldsymbol{e}_2, \boldsymbol{a} + \Delta \boldsymbol{e}_3, \boldsymbol{b} + \Delta \boldsymbol{e}_4, \boldsymbol{c} + \Delta \boldsymbol{e}_5]$. We can observe that offsets of $[\boldsymbol{a}, \boldsymbol{b}, \boldsymbol{c}]$ are different. The offsets of the former is $[\Delta \boldsymbol{e}_1, \Delta \boldsymbol{e}_2, \Delta \boldsymbol{e}_3]$, while the offsets of the latter is $[\Delta \boldsymbol{e}_3, \Delta \boldsymbol{e}_4, \Delta \boldsymbol{e}_5]$. Just adding a meaningless sequence $[\boldsymbol{t}_1, \boldsymbol{t}_2]$ that does not affect the prediction results, the token embedding offsets for $[\boldsymbol{a}, \boldsymbol{b}, \boldsymbol{c}]$ become different. These position-dependent embedding offsets make the embeddings of each token non-unique in one task, which may be detrimental to the model performance. To simulate this scenario, we design an ideal experiment where we cyclically left shift the column vectors of $\Delta \boldsymbol{E}$ and then test the model performance, as shown in Table 1. Let $\Delta \boldsymbol{E}'$ be the cyclic left shift of $\Delta \boldsymbol{E}$ by $j$ positions, defined as $\Delta \boldsymbol{E}' = [\Delta \boldsymbol{e}_{1+j}, \Delta \boldsymbol{e}_{2+j}, \cdots, \Delta \boldsymbol{e}_s, \boldsymbol{e}_1, \cdots, \boldsymbol{e}_j]$. For example, on RTE, the performance of DePT is decreased by 0.7 points after cyclically left-shifting the position-based embedding offsets by 50 positions, and increased by 0.8 points after shifting by 100 positions. On BoolQ, the performance of DePT is worse than the original after cyclically left-shifting the position-based embedding offsets. Table 1 shows that position-based embedding offsets in DePT can cause unstable prediction performance across different positions.

Table 2 reports the mean and variance of elements in input token embedding $\boldsymbol{e}$ and the elements in embedding offset $\Delta \boldsymbol{e}$ from DePT across two entire training datasets, *i.e.*, RTE and BoolQ tasks. All results are calculated by their absolute values. We can observe that the mean and variance of elements in $\Delta \boldsymbol{e}$ are only a few percent of those of elements in $\boldsymbol{e}$. For example, on the RTE task, the mean absolute value of elements in $\Delta \boldsymbol{e}$ is only 0.01, while the mean absolute value of elements in $\boldsymbol{e}$ is 6.07. This implies that DePT makes only minor changes to the input token embedding space, which may result in its inability to map the input token embeddings to the appropriate embedding space. This is because, for token $\boldsymbol{e}$, its offset can be any position $\Delta \boldsymbol{e}_i$, where $1 \leq i \leq s$. The requirement that tokens within the same task should be unique causes the offsets of DePT to become extremely small, leading to the sub-optimization of DePT. The elements in the embedding offsets of ADePT have a much larger range of values than that of DePT. The optimal embedding space may lie outside the range of DePT, whereas ADePT may be able to access this embedding space.

### 3.3  OUR METHOD: ADAPTIVE DECOMPOSED PROMPT TUNING (ADEPT)

The limitations of DePT lie in its position-based token embedding offsets. To address this issue, the input token embedding offsets should be tailored for model input, and the corresponding embedding for each token should be unique after being offset. We find that making the input token embedding offsets $\Delta \boldsymbol{E}$ a function of the input token embedding $\boldsymbol{E}$ can meet such requirements. For an input token embedding $\boldsymbol{e}$, we want to get a function $f()$, which can produce the offset of this input token embedding, namely, $f(\boldsymbol{e})$. For the input token embeddings $\boldsymbol{E} = [\boldsymbol{e}_1, \boldsymbol{e}_2, \cdots, \boldsymbol{e}_s]$, the updated input token embedding can be formulated as $\boldsymbol{E}' = \boldsymbol{E} + \Delta \boldsymbol{E} = [\boldsymbol{e}_1 + f(\boldsymbol{e}_1), \boldsymbol{e}_2 + f(\boldsymbol{e}_s), \cdots, \boldsymbol{e}_s + f(\boldsymbol{e}_s)]$. For example, the input token embeddings $[\boldsymbol{a}, \boldsymbol{b}, \boldsymbol{c}]$ can be updated as $[\boldsymbol{a} + f(\boldsymbol{a}), \boldsymbol{b} + f(\boldsymbol{b}), \boldsymbol{c} + f(\boldsymbol{c})]$. Prepending the meaningless sequence $[\boldsymbol{t}_1, \boldsymbol{t}_2]$ to input token embeddings $[\boldsymbol{a}, \boldsymbol{b}, \boldsymbol{c}]$, the updated input token embeddings are $[\boldsymbol{t}_1 + f(\boldsymbol{t}_1), \boldsymbol{t}_2 + f(\boldsymbol{t}_2), \boldsymbol{a} + f(\boldsymbol{a}), \boldsymbol{b} + f(\boldsymbol{b}), \boldsymbol{c} + f(\boldsymbol{c})]$. We can observe that the offsets for $[\boldsymbol{a}, \boldsymbol{b}, \boldsymbol{c}]$ are the same in this two scenario, *i.e.*, $[f(\boldsymbol{a}), f(\boldsymbol{b}), f(\boldsymbol{c})]$. Therefore, if such a function $f$ exists, we can achieve input token embedding offsets that are tailored for model input, and the embedding for each token is unique within a task.

To avoid increasing the number of trainable parameters, we use a shallow and narrow feed-forward neural network to approximate the function $f$. Thus, we propose Adaptive Decomposed Prompt Tuning (ADePT), which can offset the token embeddings adaptively based on the model input. We implement the shallow token-shared feed-forward neural network by a two-layer multi-layer perceptron (MLP). It consists of a down-projection matrix $\boldsymbol{W}_{\text{down}} \in \mathbb{R}^{d \times r}$ and a up-projection matrix $\boldsymbol{W}_{\text{up}} \in \mathbb{R}^{r \times d}$, and a down-projection bias $\boldsymbol{b}_1 \in \mathbb{R}^r$ and a up-projection bias $\boldsymbol{b}_2 \in \mathbb{R}^d$. Here, $r$ is the bottleneck size of the MLP. The updated input token embeddings by the shallow token-shared feed-forward neural network are formulated as:

$$\boldsymbol{E}'_i = \boldsymbol{E}_i + \text{ReLU}(\boldsymbol{E}_i \boldsymbol{W}_{\text{down}} + \boldsymbol{b}_1) \boldsymbol{W}_{\text{up}} + \boldsymbol{b}_2. \tag{3}$$

To ensure that the number of trainable parameters does not exceed that of the vanilla PT, we use a short soft prompt $\boldsymbol{P}_{s_2} \in \mathbb{R}^{l_{s_2} \times d}$, similar to DePT. The loss of ADePT is formulated as:

$$\mathcal{L}_{\mathrm{ADePT}} = -\sum_i \log P(\boldsymbol{y}_i \mid [\boldsymbol{P}_{s_2}, \boldsymbol{E}'_i] ; \Theta), \tag{4}$$

where the loss function $\mathcal{L}_{\mathrm{ADePT}}$ is optimized only with respect to the short soft prompt $\boldsymbol{P}_{s_2}$ and the parameters of the feed-forward neural network $\boldsymbol{W}_{\mathrm{down}}$, $\boldsymbol{W}_{\mathrm{up}}$, $\boldsymbol{b}_1$ and $\boldsymbol{b}_2$.

## 3.4 THEORETICAL ANALYSIS

In this section, inspired by Petrov et al. (2024), we provide a theoretical analysis towards ADePT.

The multi-head self-attention layer serves as a crucial component in each transformer layer. We analyze how PT and ADePT affect the first transformer layer. To simplify the analysis, let us consider a single head self-attention $\mathcal{H}$ in the first layer, which is parameterized by $\boldsymbol{W}_Q^{\mathcal{H}}, \boldsymbol{W}_K^{\mathcal{H}}, \boldsymbol{W}_Q^{\mathcal{H}} \in \mathbb{R}^{d \times d_{\mathcal{H}}}$. Given a input sequence embeddings $\boldsymbol{E} = (\boldsymbol{e}_1, \boldsymbol{e}_2, \ldots, \boldsymbol{e}_s) \in \mathbb{R}^{s \times d}$ with each $\boldsymbol{e} \in \mathbb{R}^d$, the output of a query vector $\boldsymbol{e}_i$ passing through the single-head self-attention $\mathcal{H}$ in the first layer is formulated as:

$$\boldsymbol{o}_i = \mathrm{Attention}\left(\boldsymbol{e}_i \boldsymbol{W}_Q^{\mathcal{H}}, \boldsymbol{E} \boldsymbol{W}_K^{\mathcal{H}}, \boldsymbol{E} \boldsymbol{W}_V^{\mathcal{H}}\right) = \mathrm{Softmax}\left(\left(\boldsymbol{e}_i \boldsymbol{W}_Q^{\mathcal{H}}\right)\left(\boldsymbol{E} \boldsymbol{W}_K^{\mathcal{H}}\right)^T\right) \boldsymbol{E} \boldsymbol{W}_V^{\mathcal{H}}, \tag{5}$$

where the scaling constant $\sqrt{d_{\mathcal{H}}}$ is ignored for notation convenience.

For the vanilla PT with the soft prompt $\boldsymbol{P} = [\boldsymbol{p}_1, \boldsymbol{p}_2, \ldots, \boldsymbol{p}_l] \in \mathbb{R}^{l \times d}$, the output of a query vector $\boldsymbol{e}_i$ passing through the single-head self-attention $\mathcal{H}$ in the first layer is formulated as:

$$\boldsymbol{o}_i^{\mathrm{PT}} = \mathrm{Attention}\left(\boldsymbol{e}_i \boldsymbol{W}_Q^{\mathcal{H}}, \mathrm{concat}[\boldsymbol{P}, \boldsymbol{E}] \boldsymbol{W}_K^{\mathcal{H}}, \mathrm{concat}[\boldsymbol{P}, \boldsymbol{E}] \boldsymbol{W}_V^{\mathcal{H}}\right)$$

$$= \underbrace{\sum_{k=1}^{l} \boldsymbol{A}_{ik} \boldsymbol{p}_k \boldsymbol{W}_V^{\mathcal{H}}}_{bias} + \underbrace{\left(1 - \sum_{k=1}^{l} \boldsymbol{A}_{ik}\right) \boldsymbol{o}_i}_{scale}, \tag{6}$$

$$\text{with} \quad \boldsymbol{A}_{ik} = \frac{\exp\left(\boldsymbol{e}_i \boldsymbol{W}_Q^{\mathcal{H}}\left(\boldsymbol{p}_k \boldsymbol{W}_K^{\mathcal{H}}\right)^T\right)}{\sum_{k=1}^{l} \exp\left(\boldsymbol{e}_i \boldsymbol{W}_Q^{\mathcal{H}}\left(\boldsymbol{p}^k \boldsymbol{W}_K^{\mathcal{H}}\right)^T\right) + \sum_{j=1}^{s} \exp\left(\boldsymbol{e}_i \boldsymbol{W}_Q^{\mathcal{H}}\left(\boldsymbol{e}_j \boldsymbol{W}_K^{\mathcal{H}}\right)^T\right)},$$

where $\boldsymbol{A}_{ik}$ is the attention score assigned to the prefix vector $\boldsymbol{p}_k$ for $\boldsymbol{e}_i$. Thus, in the first transformer layer, PT cannot affect the relative attention patterns across the content and it only scales the attention patterns down while adding a constant bias to the original output $\boldsymbol{o}_i$ (Petrov et al., 2024).

For our proposed ADePT with the soft prompt $\boldsymbol{P} = [\boldsymbol{p}_1, \boldsymbol{p}_2, \ldots, \boldsymbol{p}_l] \in \mathbb{R}^{l \times d}$ and feed-forward neural network $f$, the output of a query vector $\boldsymbol{e}_i$ passing through the single-head self-attention $\mathcal{H}$ in the first layer is formulated as:

$$\boldsymbol{o}_i^{\mathrm{ADePT}} = \mathrm{Attention}\left((\boldsymbol{e}_i + f(\boldsymbol{e}_i)) \boldsymbol{W}_Q^{\mathcal{H}}, \mathrm{concat}[\boldsymbol{P}, \boldsymbol{E} + f(\boldsymbol{E})] \boldsymbol{W}_K^{\mathcal{H}}, \mathrm{concat}[\boldsymbol{P}, \boldsymbol{E} + f(\boldsymbol{E})] \boldsymbol{W}_V^{\mathcal{H}}\right)$$

$$= \sum_{k=1}^{l} \boldsymbol{A}_{ik} \boldsymbol{p}_k \boldsymbol{W}_V^{\mathcal{H}}$$

$$+ (1 - \sum_{k=1}^{l} \boldsymbol{A}_{ik}) \mathrm{Softmax}\left(((\boldsymbol{e}_i + f(\boldsymbol{e}_i)) \boldsymbol{W}_Q^{\mathcal{H}})((\boldsymbol{E} + f(\boldsymbol{E})) \boldsymbol{W}_K^{\mathcal{H}})^T\right)(\boldsymbol{E} + f(\boldsymbol{E})) \boldsymbol{W}_V^{\mathcal{H}},$$

$$\text{with} \quad \boldsymbol{A}_{ik} = \frac{\exp\left(((\boldsymbol{e}_i + f(\boldsymbol{e}_i)) \boldsymbol{W}_Q^{\mathcal{H}})(\boldsymbol{p}_k \boldsymbol{W}_K^{\mathcal{H}})^\top\right)}{B},$$

$$B = \sum_{k=1}^{l} \exp\left(((\boldsymbol{e}_i + f(\boldsymbol{e}_i)) \boldsymbol{W}_Q^{\mathcal{H}})(\boldsymbol{p}_k \boldsymbol{W}_K^{\mathcal{H}})^T\right)$$

$$+ \sum_{j=1}^{s} \exp\left(((\boldsymbol{e}_i + f(\boldsymbol{e}_i)) \boldsymbol{W}_Q^{\mathcal{H}})((\boldsymbol{e}_j + f(\boldsymbol{e}_j)) \boldsymbol{W}_K^{\mathcal{H}})^T\right). \tag{7}$$

Hence, in the first transformer layer, ADePT can change the original relative attention patterns and add a bias dependent on the input, which makes ADePT have more expressive power than PT.

# 4 EXPERIMENTS AND RESULTS

## 4.1 EXPERIMENTAL SETUP

**Tasks and Models.** We conduct extensive experiments to validate our proposed ADePT. We consider four benchmarks and 4 other datasets: (1) GLUE (Wang et al., 2018) benchmark, which includes MNLI (Williams et al., 2018), QQP[1], QNLI (Rajpurkar et al., 2016), SST-2 (Socher et al., 2013), STS-B (Cer et al., 2017), MRPC (Dolan & Brockett, 2005), RTE (Giampiccolo et al., 2007) and CoLA (Warstadt et al., 2019); (2) SuperGLUE benchmark (Wang et al., 2019), which includes MultiRC (Khashabi et al., 2018), BoolQ (Clark et al., 2019), WiC (Pilehvar & Camacho-Collados, 2019), WSC (Levesque et al., 2012), CB (De Marneffe et al., 2019) and ReCoRD (Zhang et al., 2018); (3) MRQA 2019 Shared Task (Fisch et al., 2019), which includes Natural Questions (Kwiatkowski et al., 2019), HotpotQA (Yang et al., 2018), SearchQA (Dunn et al., 2017) and NewsQA (Trischler et al., 2017); (4) MBPP benchmark (Austin et al., 2021), which is a code generation task; (5) other datasets, which includes WinoGrande (Sakaguchi et al., 2021), Yelp-2 (Zhang et al., 2015), SciTail (Khot et al., 2018) and PAWS-Wiki (Zhang et al., 2019). Following (Asai et al., 2022; Wang et al., 2023; Shi & Lipani, 2024), we evaluate our proposed ADePT on all datasets for the T5-base model (220M) (Raffel et al., 2020), except for MBPP and ReCoRD. For the T5-3B model (Raffel et al., 2020), we focus on large and challenging datasets (*i.e.*, MNLI, ReCoRD, Natural Questions, HotpotQA, SearchQA, and NewsQA) to differentiate the performance of various PEFT methods. For the decoder-only PLMs (*i.e.*, CodeGen-350M (Nijkamp et al., 2023) and Llama3-8B (Dubey et al., 2024)), we evaluate our proposed method ADePT on MBPP benchmark.

**Baselines.** To evaluate our proposed ADePT, we compare it with five types of fine-tuning methods: (1) full fine-tuning (FT), which optimizes all the model parameters; (2) the vanilla PT (Lester et al., 2021), where target prompt vectors are initialized with randomly sampled top vocabularies; (3) the variants of PT using additional transfer or multi-task learning, including SPoT (Vu et al., 2022), ATTEMPT (Asai et al., 2022), and MPT (Wang et al., 2023); (4) the variants of PT using input token embedding offsets, *i.e.*, DePT (Shi & Lipani, 2024); (5) state-of-the-art PEFT methods including Adapter(Houlsby et al., 2019), AdapterDrop (Rücklé et al., 2021), BitFit (Ben Zaken et al., 2022), HyperFomer (Karimi Mahabadi et al., 2021), HyperDecoder (Ivison & Peters, 2022), P-tuning (Liu et al., 2021), LoRA (Hu et al., 2021), LST (Sung et al., 2022), and their multi-task learning variants.

**Implementation Details.** Following Shi & Lipani (2024), we use 100 learnable virtual tokens as the soft prompt of PT. For our proposed ADePT, we adjust the hyperparameters to maintain an equivalent number of trainable parameters as PT. For instance, in the T5-base model, the token embedding dimension $d$ is 768, so the number of trainable parameters is $l \times d = 100 \times 768 = 76800$. Following Shi & Lipani (2024), we search the length of soft prompt from 20, 40, 60, and 80. For ADePT, if using 60 virtual tokens for soft prompt, the $d_r$ is got by solving the unequal equation $60 \times 768 + 2 \times d_r \times 768 + d_r + 768 \le 76800$. Thus, the $d_r \le 19.49$ and $d$ is set to 19 because the $d$ is the integer. According to this calculation method, the corresponding $d_r$ values for soft prompt lengths of 20, 40, 60 and 80 are 39, 29, 19, and 9, respectively. For a fair comparison of the T5-base model, we directly quote performance metrics from published papers (Mahabadi et al., 2021; Karimi Mahabadi et al., 2021; Asai et al., 2022; Wang et al., 2023; Sung et al., 2022; Shi & Lipani, 2024). For T5-3B model, we consistently use 60 virtual tokens and bottleneck size $r = 19$. Due to the lack of experimental results, for a fair comparison with the T5-3B model, we reproduce the experiments of the vanilla PT and DePT. For decoder-only PLMs, following Jain et al. (2024), we use 10 virtual tokens for PT, 7 virtual tokens and rank $r_s = 3$ for DePT, 7 virtual tokens and bottleneck size $r = 1$ for ADePT, and rank 16 for LoRA. For small datasets ($< 70,000$ training samples) based on T5 model, we follow the learning strategy of Shi & Lipani (2024): we search the learning rate for the soft prompt from $3e-1$, $4e-1$, $5e-1$, and for the feed-forward neural network from $1e-4$, $1e-5$. For large datasets ($> 70,000$ training samples) based on T5 model, we use learning rate 3e-1 for the soft prompt and $1e-4$ for the feed-forward neural networks. For the MBPP benchmark, following Jain et al. (2024), we use learning rates of $1e-3$ for the prompting-style tuning method, $1e-4$ for LoRA.

---

[1] https://www.quora.com/q/quoradata/

Table 3: The experimental results on GLUE and SuperGLUE benchmarks, with the associated size of trainable parameters. All results are based on the T5-base model. We report Pearson correlation for STS-B, F1 for MultiRC (Multi), and accuracy for other tasks as test metrics.

| Method | #Para | GLUE | | | | | | | | | SuperGLUE | | | | | |
|---|---|---|---|---|---|---|---|---|---|---|---|---|---|---|---|---|
| | | MNLI | QQP | QNLI | SST-2 | STS-B | MRPC | RTE | CoLA | Mean | Multi | Bool | WiC | WSC | CB | Mean |
| *Single-Task Learning* | | | | | | | | | | | | | | | | |
| Full Finetuning[1] | 220M | 86.8 | 91.6 | 93.0 | 94.6 | 89.7 | 90.2 | 71.9 | 61.8 | 84.9 | 72.8 | 81.1 | 70.2 | 59.6 | 85.7 | 73.9 |
| Adapter[1] | 1.9M | 86.5 | 90.2 | 93.2 | 93.8 | 90.7 | 85.3 | 71.9 | 64.0 | 84.5 | 75.9 | 82.5 | 67.1 | 67.3 | 85.7 | 75.7 |
| AdapterDrop[1] | 1.1M | 86.3 | 90.2 | 93.2 | 93.6 | 91.4 | 86.3 | 71.2 | 62.7 | 84.4 | 72.9 | 82.3 | 68.3 | 67.3 | 85.7 | 75.3 |
| BitFit[1] | 280K | 85.3 | 90.1 | 93.0 | 94.2 | 90.9 | 86.8 | 67.6 | 58.2 | 83.3 | 74.5 | 79.6 | 70.0 | 59.6 | 78.6 | 72.5 |
| LoRA[2] | 3.8M | 86.3 | 89.0 | 93.2 | 94.3 | 90.9 | 90.1 | 75.5 | 63.3 | 85.3 | 72.6 | 81.3 | 68.3 | 67.3 | 92.9 | 76.5 |
| LST[2] | 3.8M | 85.6 | 88.8 | 93.3 | 94.1 | 90.7 | 90.4 | 71.9 | 58.1 | 84.1 | – | – | – | – | – | – |
| PT[4] | 76.8K | 83.4 | 90.2 | 93.1 | 91.9 | 90.2 | 90.1 | 78.8 | 60.7 | 84.8 | 65.7 | 63.7 | 50.8 | 51.9 | 67.9 | 60.0 |
| DePT[4] | 76.8K | 85.0 | 90.4 | 93.2 | 94.2 | 90.8 | 90.7 | 79.1 | 63.8 | 85.9 | 74.3 | 79.3 | 68.7 | 67.3 | 92.9 | 76.5 |
| ADePT (ours) | 76.1K | 85.7 | 90.4 | 93.2 | 94.0 | 90.9 | 91.2 | 82.0 | 65.5 | **86.6** | 74.6 | 80.2 | 68.7 | 67.3 | 96.4 | **77.4** |
| *Additional Transfer Learning or Multi-Task Learning* | | | | | | | | | | | | | | | | |
| Full Fine-tuning (m)[1] | 28M | 85.7 | 91.1 | 92.0 | 92.5 | 88.8 | 90.2 | 75.4 | 54.9 | 83.8 | 74.4 | 81.1 | 70.0 | 71.2 | 85.7 | 76.1 |
| Adapter (m)[1] | 1.8M | 86.3 | 90.5 | 93.2 | 93.0 | 89.9 | 90.2 | 70.3 | 61.5 | 84.4 | 72.6 | 82.3 | 66.5 | 67.3 | 89.3 | 75.6 |
| HyperFormer (m)[1] | 638K | 85.7 | 90.0 | 93.0 | 94.0 | 89.7 | 87.2 | 75.4 | 63.7 | 84.8 | 72.9 | 82.5 | 69.0 | 67.3 | 85.7 | 75.4 |
| HyperDecoder (m)[1] | 1.8M | 86.0 | 90.5 | 93.4 | 94.0 | 90.5 | 87.7 | 71.7 | 55.9 | 83.7 | 70.4 | 78.8 | 67.1 | 61.5 | 82.1 | 72.0 |
| SPoT (t)[1] | 76.8K | 85.4 | 90.1 | 93.0 | 93.4 | 90.0 | 79.7 | 69.8 | 57.1 | 82.3 | 74.0 | 77.2 | 67.0 | 50.0 | 46.4 | 62.9 |
| ATTEMPT (t)[1] | 232K | 84.3 | 90.3 | 93.0 | 93.2 | 89.7 | 85.7 | 73.4 | 57.4 | 83.4 | 74.4 | 78.8 | 66.8 | 53.8 | 78.6 | 70.5 |
| MPT (t)[3] | 77.6K | 85.9 | 90.3 | 93.1 | 93.8 | 90.4 | 89.1 | 79.4 | 62.4 | 85.6 | 74.8 | 79.6 | 69.0 | 67.3 | 79.8 | 74.1 |
| ATTEMPT (m)[3] | 96K | 83.8 | 90.0 | 93.1 | 93.7 | 90.8 | 86.1 | 79.9 | 64.3 | 85.2 | 74.4 | 78.5 | 66.5 | 69.2 | 82.1 | 74.1 |
| MPT (m)[3] | 10.5K | 84.3 | 90.0 | 93.0 | 93.3 | 90.4 | 89.2 | 82.7 | 63.5 | 85.8 | 74.8 | 79.2 | 70.2 | 67.3 | 89.3 | 76.1 |

[1] sourced from (Asai et al., 2022). [2] sourced from (Sung et al., 2022). [3] sourced from (Wang et al., 2023). [4] sourced from (Shi & Lipani, 2024). (m) represents additional multi-task training. (t) represents additional transfer learning.

Table 4: The experimental results on MRQA 2019 Shared Task and other datasets with the associated size of trainable parameters. All results are based on the T5-base model. We report the F1 for MRQA tasks and accuracy for other datasets as test metrics. The results are averaged over three runs and the subscripts denote standard deviation. All baseline results are quoted from (Shi & Lipani, 2024).

| Method | #Para | MRQA | | | | | Others | | | | |
|---|---|---|---|---|---|---|---|---|---|---|---|
| | | NQ | HP | SQA | News | Mean | WG | Yelp | SciTail | PAWS | Mean |
| Full Fine Tuning | 220M | 75.1 | 77.5 | 81.1 | 65.2 | 74.7 | 61.9 | 96.7 | 95.8 | 94.1 | 87.1 |
| Adapter | 1.9M | 74.2 | 77.6 | 81.4 | 65.6 | 74.7 | 59.2 | 96.9 | 94.5 | 94.3 | 86.2 |
| BitFit | 280K | 70.7 | 75.5 | 77.7 | 64.1 | 72.0 | 57.2 | 94.7 | 94.7 | 92.0 | 84.7 |
| LoRA | 3.8M | 72.4 | 62.3 | 72.5 | 56.9 | 66.0 | 58.2 | 97.1 | 94.7 | 94.0 | 86.0 |
| PT | 76.8K | 67.9 | 72.9 | 75.7 | 61.1 | 69.4 | 49.6 | 95.1 | 87.9 | 55.8 | 72.1 |
| SPoT | 76.8K | 68.2 | 74.8 | 75.3 | 58.2 | 69.1 | 50.4 | 95.4 | 91.2 | 91.1 | 82.0 |
| ATTEMPT | 232K | 70.4 | 75.2 | 77.3 | 62.8 | 71.4 | 57.6 | 96.7 | 93.1 | 92.1 | 84.9 |
| MPT | 77.6K | $72.0_{0.1}$ | $75.8_{0.1}$ | $77.2_{0.1}$ | $63.7_{0.1}$ | 72.2 | $56.5_{0.9}$ | $96.4_{0.0}$ | $95.5_{0.1}$ | $93.5_{0.1}$ | 85.5 |
| DePT | 76.8K | $73.2_{0.1}$ | $76.8_{0.3}$ | $77.6_{0.2}$ | $64.4_{0.1}$ | 73.0 | $59.0_{0.2}$ | $96.8_{0.1}$ | $95.6_{0.2}$ | $93.7_{0.1}$ | 86.3 |
| ADePT (ours) | 76.1K | $73.9_{0.0}$ | $77.1_{0.1}$ | $78.7_{0.1}$ | $64.7_{0.1}$ | 73.6 | $59.1_{0.9}$ | $96.8_{0.0}$ | $95.9_{0.3}$ | $93.7_{0.2}$ | 86.4 |

## 4.2 RESULTS BASED ON T5-BASE MODEL

**#1 Performance on GLUE and SuperGLUE benchmarks.**

As demonstrated in Table 3, our proposed ADePT surpasses leading PEFT methods, including Adapter, LoRA, BitFit, and LST, on the GLUE and SuperGLUE benchmarks, while utilizing the least trainable parameters. ADePT outperforms the vanilla PT while using comparable trainable parameters and less inference time. ADePT also outperforms the variants of PT using additional transfering learning, including SPoT, ATTEMPT and MPT while not requiring the complicated training and storage of soft prompts for source tasks. Remarkably, ADePT outperforms DePT, demonstrating that the adaptive input token embedding offsets by token-shared feed-forward neural networks are better than the position-based input token embedding offsets. Moreover, ADePT can even outperform the full finetuning method and the PEFT methods using additional multi-task learning.

**#2 Performance on MRQA 2019 Shared Task and other four datasets.**

Table 5: The experimental results on six large and challenging tasks with the associated size of trainable parameters. All results are based on the T5-3B model. We use F1 for Natural Questions, HotpotQA, SearchQA, NewsQA, and ReCoRD, and accuracy for MNLI as test metrics.

| Method | #Para | NQ | HP | SQA | News | MNLI | ReCoRD | Mean |
|---|---|---|---|---|---|---|---|---|
| LoRA | 25.8M | **80.6** | **82.6** | **87.1** | **69.5** | **91.3** | 72.8 | **80.7** |
| PT | 102.4K | 77.5 | 80.8 | 84.5 | 67.7 | 90.7 | 72.9 | 79.0 |
| DePT | 101.4K | 77.2 | 80.7 | 83.8 | 66.4 | 89.7 | 72.8 | 78.4 |
| ADePT (ours) | 101.4K | 77.7 | 80.9 | 84.7 | 67.8 | 90.9 | **73.0** | 79.2 |

Table 6: The experimental results on six large and challenging tasks with the associated size of trainable parameters. All results are based on the T5-3B model. We use EM (Exact Match) for Natural Questions, HotpotQA, SearchQA, NewsQA, and ReCoRD as test metrics.

| Method | #Para | NQ | HP | SQA | News | ReCoRD | Mean |
|---|---|---|---|---|---|---|---|
| LoRA | 25.8M | **69.7** | **67.6** | **82.5** | **55.1** | 59.2 | **66.8** |
| PT | 102.4K | 65.4 | 65.4 | 79.2 | 51.6 | 59.2 | 64.2 |
| DePT | 101.4K | 65.0 | 65.4 | 78.3 | 48.9 | 59.1 | 63.3 |
| ADePT (ours) | 101.4K | 65.8 | 65.5 | 79.4 | 51.5 | **59.3** | 64.3 |

Table 7: Performance comparison on MBPP benchmark. We report average *pass*@1 scores on CodeGen-350M and Llama3-8B models.

| Model | Method | #Para | Code Generation MBPP |
|---|---|---|---|
| CodeGen-350M | LoRA | 1.3M | **20.32** |
| | PT | 10.2K | 16.12 |
| | DePT | 10.4K | 16.83 |
| | ADePT(ours) | 10.2K | 17.86 |
| Llama3-8B | LoRA | 9.4M | **49.08** |
| | PT | 41.0K | 18.27 |
| | DePT | 42.7K | 42.50 |
| | ADePT (ours) | 41.0k | 43.22 |

Table 4 presents the performance of different PEFT methods in the MRQA dataset and four other tasks. Despite having fewer parameters (76.1K) and faster inference (shorter soft prompt), ADePT shows a significant improvement of $6.1\%$ on MRQA and $19.8\%$ on the four other datasets over the vanilla PT. Also, ADePT surpasses the variants of PT using additional transfer learning, including SPoT, ATTEMPT and MPT on MRQA and the other four tasks. Furthermore, ADePT can consistently outperform DePT on MRQA and achieve comparable performance compared to DePT on the other four tasks. Compared to Adapter, ADePT can achieve comparable performance when only using $4.0\%$ trainable parameters.

## 4.3 RESULTS BASED ON T5-3B MODEL

In this section, we evaluate our proposed ADePT the T5-3B model on six large and challenging tasks, including MNLI from the GLUE benchmark, ReCoRD from the SuperGLUE benchmark, and the MRQA 2019 Shared Task. Tables 5 and 6 present the experimental results of PT, DePT, and ADePT on six large and challenging tasks. DePT underperforms the vanilla PT across all tasks. There may be two reasons for this: (1) the position-based embedding offsets of DePT are harmful to the T5-3B model; (2) DePT is sensitive to hyperparameters, and the hyperparameters selected based on GLUE and SuperGLUE benchmarks are not conducive to the optimization of DePT. Both of these reasons indicate that the token embedding offsets based on position and the sharing of token embedding offsets among multiple tokens cause sub-optimization of PLMs, especially in billion-scale PLMs. We can observe that ADePT almost achieves the optimal results across all tasks, indicating that the use of the feed-forward neural networks to learn adaptive embedding offset tailored for each token can still map the input embedding into better embedding space on billion-scale PLMs. Although our method does not perform as well as the LoRA on the T5-3B model, our method is

Table 8: Test results of longer soft prompt lengths using the T5-base model on the GLUE benchmark.

| Method | #Para | Average Glue Performance | Inference samples per second (SST2) |
|---|---|---|---|
| PT (m=200) | 153.6K | 85.2 | 57.4 |
| DePT (m=120, r=60) | 153.6K | 86.0 | 77.2 |
| ADePT (m=120, r=39) (ours) | 152.9K | 86.5 | 72.7 |

the best among PT-style methods, and compared to LoRA, it can use significantly fewer parameters while flexibly switching parameters to adapt to different downstream tasks.

### 4.4 RESULTS BASED ON DECODER-ONLY PLMS

We evaluate our proposed ADePT on decoder-only PLMs (i.e., CodeGen-350M model (Nijkamp et al., 2023) and Llama3-8B model (Dubey et al., 2024)) through instruction tuning (Ouyang et al., 2022). We use MBPP benchmark, which is a Python program generation task (Austin et al., 2021). Following Jain et al. (2024), we use a 50-50 split for training and test. We report average *pass@1* scores to evaluate the performance, as shown in Table 7. We can observe that ADePT performs best among PT-style methods, demonstrating its effectiveness. In the Llama3-8B model, the vanilla PT performs much worse than DePT and ADePT. This shows that the inability of PT to change the relative attention patterns limits its adaptation ability, whereas DePT and ADePT perform better because they can change the relative attention patterns. Also, in CodeGen-350M and Llama3-8B, the use of adaptive token embedding offsets helps ADePT perform better than DePT. Although ADePT performs slightly worse than LoRA on the MBPP benchmark, ADePT requires far fewer parameters than LoRA. More importantly, LoRA needs to merge weights, which typically limits it to only a single downstream task. In contrast, ADePT can flexibly switch between tasks and adapt to multiple downstream tasks simultaneously, which is a unique advantage of PT-style methods.

### 4.5 FURTHER ANALYSIS

**Performance using longer soft prompt.** Table 8 compares the performance of vanilla PT, DePT, and ADePT under a comparable number of trainable parameters (corresponding to vanilla PT with a length of 200). The results show that ADePT outperforms both vanilla PT and DePT, demonstrating the effectiveness of its adaptive embedding offsets. In terms of inference speed, ADePT achieves speeds that exceed those of PT and match those of DePT. Although the additional shallow token-shared feed-forward neural network introduces some latency, the impact is minimal. It is worth noting that ADePT requires computing embedding offsets for each token in real-time for every model input, whereas DePT relies on fixed token embedding offsets. This real-time computation may introduce additional latency; however, if the embedding offsets obtained from the ADePT method are precomputed and added to the corresponding tokens in advance, this latency can be eliminated. Furthermore, since the addition operation $\mathbf{E} + \mathbf{AB}$ in DePT cannot be avoided, the theoretical upper-speed limit of ADePT is expected to be faster than that of DePT, highlighting its potential to achieve both superior performance and faster inference speeds.

### 5 CONCLUSION

We propose a new parameter-efficient fine-tuning method, *i.e.*, Adaptive Decomposed Prompt Tuning (ADePT), which consists of a short soft prompt and a shallow token-shared feed-forward neural network. The feed-forward neural network can learn a unique offset for each input token and map the input token embeddings into a better embedding space in a position-independent manner. Extensive experiments demonstrate that ADePT outperforms leading PEFT methods, including full fine-tuning, in certain scenarios. Further analysis demonstrates that ADePT exhibits a faster inference speed compared to the vanilla PT with a comparable number of parameters, while simultaneously achieving superior performance. We provide a theoretical analysis demonstrating that ADePT exhibits greater expressive power compared to PT and DePT.

ACKNOWLEDGMENTS

This research was supported by National Natural Science Foundation of China (No.62476277), National Key Research and Development Program of China (NO.2024YFE0203200), CCF-ALIMAMA TECH Kangaroo Fund (No.CCF-ALIMAMA OF 2024008), and Huawei-Renmin University joint program on Information Retrieval. We also acknowledge the support provided by the fund for building worldclass universities (disciplines) of Renmin University of China and by the funds from Beijing Key Laboratory of Big Data Management and Analysis Methods, Gaoling School of Artificial Intelligence, Renmin University of China, from Engineering Research Center of Next-Generation Intelligent Search and Recommendation, Ministry of Education, from Intelligent Social Governance Interdisciplinary Platform, Major Innovation & Planning Interdisciplinary Platform for the "DoubleFirst Class" Initiative, Renmin University of China, from Public Policy and Decision-making Research Lab of Renmin University of China, and from Public Computing Cloud, Renmin University of China.

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

## APPENDIX OVERVIEW

The appendix is structured as follows:

**Appendix A**   provides a theoretical analysis towards DePT. We combine the theoretical analysis from Section 3.4 and the Appendix A to theoretically explain why ADePT performs better than DePT.

**Appendix B**   provides additional experiments to further analyze our proposed ADePT.

**Appendix C**   provides a more detailed implementation of our experiments.

**Appendix D**   provides a detailed description of datasets.

**Appendix E**   provides detailed hyperparameters of our experiments.

## A   HOW DECOMPOSED PROMPT TUNING AFFECT THE FIRST MULTI-HEAD SELF-ATTENTION LAYER?

In this section, we analyze how the DePT affects the first multi-head self-attention layer.

Given the soft prompt $\boldsymbol{P} = [\boldsymbol{p}_1, \boldsymbol{p}_2, \ldots, \boldsymbol{p}_s] \in \mathbb{R}^{l \times d}$ and the offset embeddings $\Delta \boldsymbol{E} = \boldsymbol{A}\boldsymbol{B} \in \mathbb{R}^{s \times d}$, the output of a query vector $\boldsymbol{x}_i$ passing through the single-head self-attention $\mathcal{H}$ is formulated as,

$$
\begin{aligned}
\boldsymbol{o}_i^{\text{DePT}} &= \text{Attention}\left(\left(\boldsymbol{e}_i + \Delta\boldsymbol{e}_i\right)\boldsymbol{W}_Q^{\mathcal{H}}, \text{concat}[\boldsymbol{P}, \boldsymbol{E} + \Delta\boldsymbol{E}]\boldsymbol{W}_K^{\mathcal{H}}, \text{concat}[\boldsymbol{P}, \boldsymbol{E} + \Delta\boldsymbol{E}]\boldsymbol{W}_V^{\mathcal{H}}\right), \\
&= \sum_{k=1}^{l} A_{ik}\boldsymbol{p}_k\boldsymbol{W}_V^{\mathcal{H}} + \left(1 - \sum_{k=1}^{l} A_{ik}\right)\text{Softmax}\left(\left(\left(\boldsymbol{e}_i + \Delta\boldsymbol{e}_i\right)\boldsymbol{W}_Q^{\mathcal{H}}\right)\left(\left(\boldsymbol{E} + \Delta\boldsymbol{E}\right)\boldsymbol{W}_K^{\mathcal{H}}\right)^T\right)\Delta\boldsymbol{E}\boldsymbol{W}_V^{\mathcal{H}} \\
&\quad + (1 - \sum_{k=1}^{l} A_{ik})\text{Softmax}\left(\left(\left(\boldsymbol{e}_i + \Delta\boldsymbol{e}_i\right)\boldsymbol{W}_Q^{\mathcal{H}}\right)\left(\left(\boldsymbol{E} + \Delta\boldsymbol{E}\right)\boldsymbol{W}_K^{\mathcal{H}}\right)^T\right)\boldsymbol{E}\boldsymbol{W}_V^{\mathcal{H}}, \\
\text{with} \quad A_{ik} &= \frac{\exp\left(\left(\left(\boldsymbol{e}_i + \Delta\boldsymbol{e}_i\right)\boldsymbol{W}_Q^{\mathcal{H}}\right)\left(\boldsymbol{p}_k\boldsymbol{W}_K^{\mathcal{H}}\right)^{\top}\right)}{C}, \\
C &= \sum_{k=1}^{l} \exp\left(\left(\left(\boldsymbol{e}_i + \Delta\boldsymbol{e}_i\right)\boldsymbol{W}_Q^{\mathcal{H}}\right)\left(\boldsymbol{p}_k\boldsymbol{W}_K^{\mathcal{H}}\right)^{\top}\right) \\
&\quad + \sum_{j=1}^{s} \exp\left(\left(\left(\boldsymbol{e}_i + \Delta\boldsymbol{e}_i\right)\boldsymbol{W}_Q^{\mathcal{H}}\right)\left(\left(\boldsymbol{e}_j + \Delta\boldsymbol{e}_j\right)\boldsymbol{W}_K^{\mathcal{H}}\right)^{\top}\right),
\end{aligned}
\tag{8}
$$

where $A_{ik}$ is the attention score gives to the prefix vector $\boldsymbol{p}_k$ for a given query vector $\boldsymbol{e}_i$. We can observe that, in the first transformer layer, DePT can change the relative attention patterns. However, compared to ADePT, the attention patterns change along the change of position. Also, DePT cannot add a bias dependent on model input in the first transformer layer.

# B ADDITIONAL EXPERIMENTS

## B.1 ABLATION STUDY

Table 9: The comparison of training time based on T5-3B model. "h" means hours.

| Method | #Para | NQ | HP |
|---|---|---|---|
| LoRA | 25.8M | 9.73 h | 9.63 h |
| PT | 153.6K | 12.60 h | 12.53 h |
| DePT | 153.6K | 12.60 h | 12.53 h |
| ADePT (ours) | 152.9K | 12.67 h | 12.58 h |

Table 10: the experimental results of fine-tuning both the prompt and the embedding matrix, as well as our proposed ADePT based on the T5-base model for RTE.

| Method | #Para | RTE |
|---|---|---|
| Finetuning prompt and embedding matrix | 24.8M | 76.3 |
| ADePT (ours) | 76.1K | 82.0 |

Table 11: the experimental results of ADePT/DePT when used without the token offsets but only learned soft prompt. The results are based on the T5-base model and the RTE task.

| Method | With token offsets | Without token offsets |
|---|---|---|
| DePT | 79.1 | 78.4 |
| ADePT (ours) | 82.0 | 58.3 |

Table 12: The results of ADePT with different bottleneck sizes using the T5-base model on the RTE.

| The size of the Bottleneck | 5 | 10 | 20 | 30 |
|---|---|---|---|---|
| ADePT (ours) | 78.4 | 82.0 | 82.0 | 79.9 |

Table 13: The results of ADePT with soft prompt lengths using the T5-base model on the RTE.

| The length of soft prompt | 20 | 40 | 50 | 60 | 70 | 80 |
|---|---|---|---|---|---|---|
| ADePT (ours) | 72.7 | 77.7 | 79.1 | 82.0 | 80.6 | 79.1 |

Table 14: The comparison of ADePT with only feed-forward neural network and the original ADePT based on T5-base model for the RTE.

| Method | #Para | RTE |
|---|---|---|
| Only Feed-forward Neural Network | 76.1K | 73.4 |
| ADePT (ours) | 76.1K | 82.0 |

Table 9 indicates that PT, DePT, and ADePT need similar training time. The PT-family method needs longer training time than LoRA due to the longer input sequence.

Table 10 shows the experimental results of fine-tuning both the prompt and the embedding matrix, as well as our proposed ADePT. We use the same length soft prompt, and we search learning rates for prompt matrix from {3e-1, 4e-1, 5e-1} and embedding matrix from {1e-3, 1e-4, 1e-5}. We observe that fine-tuning both the prompt and the embedding matrix underperforms our proposed ADePT. Additionally, this method requires fine-tuning a large number of parameters, which may lead to overfitting.

Table 11 show the experimental results of ADePT/DePT when used without the token offsets but only learned soft prompt. We can observe that the token offsets of ADePT play a much more important role than DePT.

Table 12 presents how the size of the bottleneck affects the performance of the RTE task. We can observe that an overly small or overly large bottleneck size will cause a decline in performance. When it is too small, it can lead to under-fitting; when it is too large, it can lead to over-fitting.

Table 13 presents how the length of the prompt affects the performance on the RTE task. We can observe that performance drops significantly when the prompt length is less than 40. Performance is optimal when the prompt length is 50 or 60, but it decreases when the prompt length is too large, possibly due to overfitting.

Table 14 shows that the performance on the RTE task when all parameters are relocated to the learnable projection (prompt length = 0, bottleneck size = 49, trainable parameters = 76.1k) is 73.4, indicating the soft prompt is necessary.

## B.2 FEW-SHOT LEARNING

Table 15: Few-shot learning results, obtained from three runs, with $k = \{4, 16, 32\}$ training samples on the BooQ, CB and SciTail datasets. Baseline results are directly quoted from Shi & Lipani (2024).

| Task | $k$-shot #Para | Full Finetuning 220M | AD 1.9M | PT 76.8K | ST 76.8K | HF 638K | $(IA)^3$ 55.3K | ATP 232K | MPT 77.6K | DePT 76.8K | ADePT (ours) 76.1K |
|---|---|---|---|---|---|---|---|---|---|---|---|
| BoolQ | 4 | 50.5 | 53.4 | 61.6 | 50.5 | 48.0 | 56.7 | 61.8 | 62.2 | $62.7_{5.4}$ | $68.7_{0.4}$ |
| | 16 | 56.5 | 51.4 | 61.9 | 50.6 | 50.2 | 62.0 | 60.0 | 63.3 | $66.9_{4.4}$ | $69.9_{1.3}$ |
| | 32 | 58.4 | 54.5 | 61.7 | 61.2 | 58.3 | 67.2 | 65.3 | 68.9 | $67.2_{3.4}$ | $70.0_{1.2}$ |
| CB | 4 | 57.7 | 51.1 | 53.5 | 71.4 | 60.7 | 65.5 | 67.9 | 73.6 | $75.0_{5.1}$ | $32.1_{2.6}$ |
| | 16 | 77.0 | 74.8 | 63.5 | 64.3 | 76.3 | 71.4 | 71.4 | 78.6 | $78.6_{4.3}$ | $36.7_{2.3}$ |
| | 32 | 80.0 | 74.8 | 67.8 | 64.3 | 81.4 | 75.0 | 78.5 | 82.1 | $82.1_{2.3}$ | $39.5_{3.1}$ |
| SciTail | 4 | 79.6 | 79.5 | 57.7 | 69.6 | 82.0 | 65.4 | 80.2 | 80.2 | $78.1_{2.5}$ | $76.9_{3.2}$ |
| | 16 | 80.0 | 83.2 | 60.8 | 71.9 | 86.5 | 74.4 | 79.5 | 87.3 | $78.5_{1.4}$ | $82.1_{2.0}$ |
| | 32 | 81.9 | 85.0 | 60.2 | 71.9 | 85.8 | 80.4 | 80.2 | 86.3 | $85.4_{3.1}$ | $82.6_{2.6}$ |

Table 15 shows the few-shot learning results with $k = \{4, 16, 32\}$ training samples on BoolQ, CB and SciTail datasets. We pre-train both the soft prompt and the feed-forward neural network on source tasks and select the best checkpoint to initialize the parameters. We can observe that ADePT performs best on the BoolQ dataset, performs well on the SciTail dataset, and performs the worst on the CB dataset. This might indicate that ADePT is unsuitable for few-shot learning, which is reasonable since learning the embedding offsets for each token using a feed-forward neural network requires considerable training samples.

## B.3 UNSUCCESSFUL ATTEMPTS

We attempted to perform instruction tuning on a larger scale of the GSM8K dataset (Cobbe et al., 2021) using Llama3 8B, with 7,473 examples in the training set and 1,319 examples in the test set. The goal of this instruction tuning was to fine-tune the model to better solve mathematical reasoning problems in the GSM8K dataset. However, unfortunately, we were not successful in our training attempts, including both ADePT and DePT. This might be because the simple low-rank matrix multiplication and shallow neural networks are unable to fit the input token embedding offsets required by large datasets when working with large models that have a very large vocabulary. Addressing this issue will be a direction for our future research.

## C  ADDITIONAL IMPLEMENTATION DETAILS

We implement our experiments by using Pytorch[2], Huggingface Transformers[3], and Huggingface PEFT [4]. We evaluate our proposed ADePT in four PLMs, *i.e.*, T5-base model[5], T5-3B model[6], CodeGen-350M[7] and Llama3-8B[8]. Following Asai et al. (2022); Wang et al. (2023); Shi & Lipani (2024), we train the T5 model using the original checkpoint rather than the LM-adapted 1.1 version (Lester et al., 2021). For the T5-3B model, due to the limitations of computational resources, we select the several most convincing datasets to evaluate our proposed ADePT. We think that convincing datasets should have a large training dataset and large test dataset, and be challenging for the T5-base model. We use the criteria of more than 70,000 training samples, an accuracy/F1 score of less than 90% on the T5-base model, and more than 4,000 test samples to select the datasets to test our proposed ADePT on the T5-3B model. Shi & Lipani (2024) found that training PT for additional steps typically leads to performance improvements, and we follow this setting to train our proposed ADePT. We measure the latency of ADePT by running a feed-forward neural network for each token in real time.

For the small datasets ($< 70,000$ training samples), following Shi & Lipani (2024), we search the learning rates for the soft prompt from $\{3e-1, 4e-1, 5e-1\}$ and for the feed-forward neural network from $\{1e-4, 1e-5\}$. We also search for the prompt length from $\{20, 40, 60, 80\}$ with corresponding bottleneck sizes of $\{39, 29, 19, 9\}$ to ensure the number of trainable parameters remains below 76.8K. For the large datasets ($> 70,000$ training samples), we set the prompt length as 60, the bottleneck size as 19, the prompt learning rate as $3e-1$, and the feed-forward neural network learning rate as $1e-4$. For decoder-only PLMs, following Jain et al. (2024), we use 10 virtual tokens for PT, 7 virtual tokens and rank $r_s = 3$ for DePT, 7 virtual tokens and bottleneck size $r = 1$ for ADePT, and rank 16 for LoRA. For the MBPP benchmark, following Jain et al. (2024), we use learning rates of $1e-3$ for the prompting-style tuning method and $1e-4$ for LoRA.

Following Shi & Lipani (2024), for the T5-base model and the small datasets, we train the model for 30,000 steps; for the T5-base model and the large datasets, we train the model for 300,000 steps. For the T5-3B model, we train the model for 30,000 steps. In each trial of the t5-base model and the T5-3B model, we evaluate the performance every 1,000 steps and select the best checkpoint based on the optimal performance on the evaluation set. For the MBPP benchmarks, following Jain et al. (2024), we train the model for 10 epochs. We train the model with a batch size of 32, except for the MBPP benchmark, where we use a batch size of 4. We typically use a maximum sequence length of 256, except for SuperGLUE-MultiRC, where the maximum sequence length is 348, and MRQA, where it is 512.

For the few-shot learning, following the prior works (Asai et al., 2022; Wang et al., 2023; Shi & Lipani, 2024), we first pre-train five source tasks (*i.e.*, MNLI, QQP, SST-2, SQUAD, and ReCoRD), and then select the best checkpoint to use as the initialization for few-shot training.

---

[2]https://pytorch.org/
[3]https://github.com/huggingface/transformers
[4]https://github.com/huggingface/peft
[5]https://huggingface.co/google-t5/t5-base
[6]https://huggingface.co/google-t5/t5-3b
[7]https://huggingface.co/Salesforce/codegen-350M-mono
[8]https://huggingface.co/meta-llama/Meta-Llama-3-8B

# D  DATASETS

Table 16: The datasets assessed in this study are described as follows. The term "Source Length" refers to the average length of source sentences in the training set, whereas "Target Length" indicates the average length of target sentences in the training set. The term "Train" refers to the number of samples in the training set, whereas "Valid" and "Test" indicate the number of samples in the validation set and test set, respectively. The term "Type" refers to the task type of the dataset.

| Dataset | Source Length | Target Length | Train | Valid | Test | Type |
|---|---|---|---|---|---|---|
| *GLUE Benchmark* | | | | | | |
| MNLI | 31.8 | 1.0 | 392,702 | 9,832 | 9,815 | NLI |
| QQP | 24.1 | 1.0 | 362,846 | 1,000 | 40,431 | Paraphrase |
| QNLI | 38.4 | 1.0 | 103,743 | 1,000 | 5,463 | NLI |
| SST-2 | 10.4 | 1.0 | 66,349 | 1,000 | 872 | Sentiment |
| STS-B | 21.9 | 1.0 | 5,749 | 750 | 750 | Sent. Similarity |
| MRPC | 45.9 | 1.0 | 3,668 | 204 | 204 | Paraphrase |
| RTE | 54.4 | 1.0 | 2,490 | 138 | 139 | NLI |
| CoLA | 8.7 | 1.0 | 8,551 | 521 | 522 | Acceptability |

| Dataset | Source | Target | Train | Valid | Test | Type |
|---|---|---|---|---|---|---|
| *SuperGLUE Benchmark* | | | | | | |
| MultiRC | 286.1 | 1.0 | 27,243 | 2,424 | 2,424 | Question Answering |
| BoolQ | 108.3 | 1.0 | 9,427 | 1,635 | 1,635 | Question Answering |
| WiC | 18.4 | 1.0 | 5,428 | 319 | 319 | Word Sense Disambiguation |
| WSC | 28.1 | 1.0 | 554 | 52 | 52 | Common Sense Reasoning |
| CB | 64.6 | 1.0 | 250 | 28 | 28 | NLI |
| ReCoRD | 210.7 | 1.5 | 137,484 | 1,370 | 15,176 | Common Sense Reasoning |

| Dataset | Source | Target | Train | Valid | Test | Type |
|---|---|---|---|---|---|---|
| *MRQA 2019 Shared Task* | | | | | | |
| NaturalQuestions | 242.7 | 4.5 | 103,071 | 1,000 | 12836 | Question Answering |
| HotpotQA | 225.7 | 2.6 | 71,928 | 1,000 | 5,901 | Question Answering |
| SearchQA | 942.8 | 2.0 | 116,384 | 1,000 | 16,980 | Question Answering |
| NewsQA | 615.5 | 5.1 | 73,160 | 1,000 | 4,212 | Question Answering |

| Dataset | Source | Target | Train | Valid | Test | Type |
|---|---|---|---|---|---|---|
| *Other Datasets* | | | | | | |
| WinoGrande | 23.8 | 1.0 | 39,398 | 1,000 | 1,267 | Common Sense Reasoning |
| YelpPolarity | 134.0 | 1.0 | 100,000 | 1,000 | 38,000 | Sentiment |
| SciTail | 30.8 | 1.0 | 23,596 | 652 | 652 | NLI |
| PAWS | 44.7 | 1.0 | 4,9401 | 8,000 | 8,000 | Sent. Similarity |

# E HYPERPARAMETERS

Table 17: Hyperparameters of small datasets for ADePT on T5-base model.

| Hyperparameter | Assignment |
| --- | --- |
| number of steps | 30,000 steps (evaluate every 1,000 steps) |
| batch size | 32 |
| maximum learning rate ($\alpha_1$) | 3e-1, 4e-1, 5e-1 |
| maximum learning rate ($\alpha_2$) | 1e-4, 1e-5 |
| length of the soft prompt ($m$) | 20, 40, 60, 80 |
| maximum sequence length | 256 |
| learning rate optimizer | AdamW |
| Adam epsilon | 1e-6 |
| Adam beta weights | 0.9, 0.98 |
| learning rate scheduler | Warmup linear |
| Weight decay | 0.01 |
| Warmup steps | 500 |

Table 18: Hyperparameters of large datasets for ADePT on T5-base model.

| Hyperparameter | Assignment |
| --- | --- |
| number of steps | 300,000 steps (evaluate every 1,000 steps) |
| batch size | 16 |
| gradient accumulation steps | 2 |
| maximum learning rate ($\alpha_1$) | 3e-1 |
| maximum learning rate ($\alpha_2$) | 1e-4 |
| length of the soft prompt ($m$) | 60 |
| maximum sequence length | 512 |
| learning rate optimizer | AdamW |
| Adam epsilon | 1e-6 |
| Adam beta weights | 0.9, 0.98 |
| learning rate scheduler | Warmup linear |
| Weight decay | 0.01 |
| Warmup steps | 500 |

Table 19: Hyperparameters for ADePT on T5-3B model.

| Hyperparameter | Assignment |
|---|---|
| number of steps | 30,000 steps (evaluate every 1,000 steps) |
| batch size | 16 |
| gradient accumulation steps | 2 |
| maximum learning rate ($\alpha_1$) | 3e-1 |
| maximum learning rate ($\alpha_2$) | 1e-4 |
| length of the soft prompt ($m$) | 60 |
| maximum sequence length | 512 |
| learning rate optimizer | AdamW |
| Adam epsilon | 1e-6 |
| Adam beta weights | 0.9, 0.98 |
| learning rate scheduler | Warmup linear |
| Weight decay | 0.01 |
| Warmup steps | 500 |

