# OpenReview forum: "ADePT: Adaptive Decomposed Prompt Tuning for Parameter-Efficient Fine-tuning"
_ICLR.cc/2025/Conference — ICLR 2025 Poster_

### Official Review · Reviewer_Z8Tq · 2024-10-28

**Soundness:** 3
**Presentation:** 3
**Contribution:** 2
**Rating:** 6
**Confidence:** 3

**Summary:**

This paper is about prompt tuning, which improves on the DePT method and propose a method called ADePT. It replaces fixed token embedding offsets with a shallow feed-forward neural network that can dynamically generate offsets for the input token embeddings, thereby providing better adaptability to different input tokens.

**Strengths:**

1. The introduction of a token-shared feed-forward neural network as a solution is well-motivated, by effectively identifying critical limitations in the DePT method of the static token embedding offsets.
2. The experiments are thorough, the performance are reported in many datasets, including NLU and NLG.

**Weaknesses:**

1. Lack of analysis of the proposed method. For example, the method includes the bottleneck hyperparameter that controls the total #params that does not exceed vanilla PT. However, the size of bottleneck as well as the corresponding prompt length on the performance is not clear. It is also interesting to show if relocated all the #params on the learnable projection, where prompt length=0.
2. In Table 8, comparing with the improvement DePT brings to PT in both accuracy and latency, the proposed ADePT seems trade latency to accuracy. Moreover, the complexity introduced in the learnable projection hurts the few-shot performance compared with DePT.
3. The method is only tested on two T5 language models, including the base and 3B variants. It should also be tested on the decoder models.

**Questions:**

See weaknesses.

---

> ### Author Response · Authors · 2024-11-21
> **Response to Reviewer Z8Tq (Part 1/2)**
>
> Dear Reviewer Z8Tq,
>
> We sincerely thank you for your time and effort in evaluating our manuscript. Your thorough evaluation and insightful comments are highly appreciated. We will address each of your questions point by point and hope to resolve your concerns effectively.
>
> ### _**For weakness 1**: Lack of analysis of the proposed method. For example, the method includes the bottleneck hyperparameter that controls the total #params that does not exceed vanilla PT. However, the size of bottleneck as well as the corresponding prompt length on the performance is not clear. It is also interesting to show if relocated all the #params on the learnable projection, where prompt length=0._
>
> **A1**: Thank you for your valuable suggestion.
>
> **First**, we present how the size of the bottleneck affects the performance of RTE task, as presented below and in Table 13 of Appendix B. We can observe that an overly small or overly large bottleneck size will cause a decline in performance. When it is too small, it can lead to under-fitting; when it is too large, it can lead to over-fitting.
>
> | The size of the Bottleneck | 5  | 10  | 20 | 30 |
> | --- | --- | --- | --- | --- |
> | Performance | 78.4 | 82.0 | 82.0 | 79.9 |
>
> **Second**, we present how the length of the prompt affects the performance on RTE task, as presented below and in Table 14 of Appendix B. We can observe that performance drops significantly when the prompt length is less than 40. Performance is optimal when the prompt length is 50 or 60, but it decreases when the prompt length is too large, possibly due to overfitting.
>
> | The size of the Bottleneck | 20 | 40 | 50 | 60 | 70 | 100 |
> | --- | --- | --- | --- | --- | --- | --- |
> | Performance | 72.7 | 77.7 | 79.1 | 82.0 | 80.6 | 79.1 |
>
> **Third**, as presented below and in Table 15 of Appendix B, the performance on RTE task when all parameters are relocated to the learnable projection (prompt length = 0, bottleneck size = 49, #params = 76.1k) is 73.4, indicating the soft prompt is necessary.
>
> | Method | #Para  | RTE|
> | --- | --- | --- |
> | Only Feed-forward Neural Network | 76.1K| 73.4 |
> |ADePT (ours) | 76.1K |82.0 |
>
> We are working on adding the experimental results of other datasets to strengthen the above conclusions.
>
> ### _**For weakness 2**: In Table 8, comparing with the improvement DePT brings to PT in both accuracy and latency, the proposed ADePT seems trade latency to accuracy. Moreover, the complexity introduced in the learnable projection hurts the few-shot performance compared with DePT._
>
> **A2**: Thank you for your valuable question. Because our proposed ADePT needs to compute the embedding offsets for each token on each model input in real-time, whereas the token embedding offsets in DePT are fixed, ADePT may introduce some latency. However, if we precompute and add the embedding offsets obtained from the ADePT method to the corresponding tokens in advance, this latency will be eliminated. Furthermore, since the addition operation $E + AB$ in DePT cannot be avoided, the theoretical upper-speed limit of ADePT would be faster than that of DePT.
>
> For the few-shot performance, in the future, we will explore how to improve the few-shot performance for ADePT.

---

> > ### Author Response · Authors · 2024-11-21
> > **Response to Reviewer Z8Tq (Part 2/2)**
> >
> > ### _**For weakness 3**: The method is only tested on two T5 language models, including the base and 3B variants. It should also be tested on the decoder models._
> >
> > **A3**: Thank you for your valuable suggestion. We conduct experiments on the MBPP benchmark[1] to test our proposed ADePT on two decoder-only large language models, namely, CodeGen-350M [2] and Llama3-8B [3]. The implementation details are also provided in the revised manuscript. The experimental results are presented as follows, and also in Table 7 in the revised manuscript. We can observe that our proposed ADePT consistently outperforms the vanilla PT and DePT, indicating our proposed method can scale to decoder-only PLMs and larger PLMs.
> >
> > | Model         | Method    | #Para      | MBPP|
> > |---------------|-----------|------------|----------------------|
> > | **CodeGen-350M** | LoRA      | 1.3M       | 20.32                |
> > |               | PT        | 10.2K      | 16.12                |
> > |               | DePT      | 10.4K      | 16.83                |
> > |               | ADePT(ours) | 10.2K      | 17.86                |
> > | **Llama3-8B** | LoRA      | 9.4M       | 49.08                |
> > |               | PT        | 41.0K      | 18.27                |
> > |               | DePT      | 42.7K      | 42.50                |
> > |               | ADePT (ours) | 41.0k      | 43.22                |
> >
> > [1] Jacob Austin, Augustus Odena, Maxwell Nye, Maarten Bosma, Henryk Michalewski, David Dohan, Ellen Jiang, Carrie Cai, Michael Terry, Quoc Le, et al. Program synthesis with large language models. arXiv preprint arXiv:2108.07732, 2021
> >
> > [2] Erik Nijkamp, Bo Pang, Hiroaki Hayashi, Lifu Tu, Huan Wang, Yingbo Zhou, Silvio Savarese, and Caiming Xiong. Codegen: An open large language model for code with multi-turn program synthesis. In ICLR 2023.
> >
> > [3] Abhimanyu Dubey, Abhinav Jauhri, Abhinav Pandey, Abhishek Kadian, Ahmad Al-Dahle, Aiesha Letman, Akhil Mathur, Alan Schelten, Amy Yang, Angela Fan, et al. The llama 3 herd of models. arXiv preprint arXiv:2407.21783, 2024.

---

> > > ### Comment · Reviewer_Z8Tq · 2024-11-21
> > >
> > > Thank you for the response and the additional experiments. I have raised my scores.

---

> ### Author Response · Authors · 2024-11-23
> **Thanks for the reply**
>
> Dear Reviewer Z8Tq,
>
> We sincerely thank you for your recognition of this paper and your valuable suggestions.
>
> Thank you for raising your review score. We feel very encouraged.
>
> Best regards,
>
> Authors

---

### Official Review · Reviewer_cKVM · 2024-10-29

**Soundness:** 4
**Presentation:** 3
**Contribution:** 4
**Rating:** 8
**Confidence:** 5

**Summary:**

This paper proposed to extend Prompt Tuning and DePT, by replacing the position-based updates that DePT applies to the true token embeddings with content-based updates by passing the token embedding through a small (bottlenecked, down-project + up-project) MLP and adding that to the original token embedding.

They test its effectiveness on many datasets, against many baselines, and use multiple frozen models. They find that their method performs best in many cases.

**Strengths:**

The paper introduces the other work in the space really well and does a good job contextualizing itself among that work.

The pilot experiments highlighting the weakness of DePT are a good motivation for their work.

The paper compares to a lot of different baselines, including PEFT methods beyond just prompt tuning.

The paper evaluates the method on a lot of different datasets, increasing the trust you can put into setting good results if you used it on your task.

The paper uses multiple different pre-trained models as the frozen models the PEFTs are applied to. These are also at two different scales. It is good to see the results still hold.

**Weaknesses:**

The weakness of DePT is outlined in the paper as its "fixed token embedding offsets". This point would be much clearer if it was re-framed as the DePT offsets are "position-based" while the ADePT offsets are "content/token-based". Both are "fixed token embedding offsets" (ADePT output is fixed once the input token is know, it isn't contextual). This framing would make a lot of their examples about the issues much clearer. For example the section about the [t1, t2] being added causing a shift if which offsets are applied where much clearer. It also makes their point about the DePT offsets not doing much because they have to handle all tokens more obvious! It also could make this example much clearer where it could cast [t1,t2] as a "system prompt" added after the fact that messes up the learned position embeddings from DePT (and also hightlight how DePT may not play nicely with prompt engineering which ADePT probably would).

They state that "embedding for each token should be unique after being offset" as a critique of DePT, but thinking of DePT as position-wise offsets it doesn't seem like a problem, especially given positional embeddings work.

The prose can be tighten up quite a bit. There are lots of parts that repeat themselves multiple times, for example the position-wise implementation of DePT is over-explained multiple times. Similarly, much of the algebraic manipulations of the parameter counts could be omitted.

Many of the increases in performance are rather small, although the simplicity of this method makes that more acceptable.

It is unclear is they use the optimization stabilization of Razdaibiedina et al 2023 they mention in the introduction when using PT.

Much of the baseline performance numbers are from other works, opening the possibility of a mis-match in setting. For example, the numbers for SPoT in Table 3 are surprising as they are lower than the Prompt Tuning numbers, but they are taken from different papers.

**Questions:**

Did you compare to fine-tuning the prompt and the embedding matrix? The would be a small loss in generality (tokens not seen during training would not get updated) but it would be a much simpler implementation. I would be curious to see how this version performed on CB too, as a possible explanation of ADePT's poor performance could be that the NN offsets don't work well on unattested words.

When measuring the latency of different methods, where the token offsets for ADePT precomputed and folded into the embedding table or was the NN run for each token?

Did you try using the LM-adapted model from Lester et al 2021 instead of the span-corrupted T5 models used here? It is also unclear if you used the original T5 models or the T5 1.1 models, IIRC, some of the datasets used were seen during pre-training of the original T5 models.

As the DePT offsets only differ between each position, it seems reasonable that they have low mean and variance. In contrast, ADePT is per-token and content based so it seems reasonable that it would have a much higher variance. I'm not sure mean and variance is the correct metric to gauge how much these offsets are actually doing (or if they are sub-optimized). Something like a norm of the offsets might make more sense? Or it could have been measured directly in an ablation where the soft prompt learned in ADePT/DePT is used without the token offsets.

---

> ### Author Response · Authors · 2024-11-21
> **Response to Reviewer cKVM (Part 1/4)**
>
> Dear Reviewer cKVM,
>
> We sincerely thank you for your time and effort in evaluating our manuscript. Your thorough evaluation and insightful comments are highly appreciated. We will address each of your questions point by point and hope to resolve your concerns effectively.
>
> ### _**For weakness1**: The weakness of DePT is outlined in the paper as its "fixed token embedding offsets". This point would be much clearer if it was re-framed as the DePT offsets are "position-based" while the ADePT offsets are "content/token-based". Both are "fixed token embedding offsets" (ADePT output is fixed once the input token is know, it isn't contextual). This framing would make a lot of their examples about the issues much clearer. For example the section about the [t1, t2] being added causing a shift if which offsets are applied where much clearer. It also makes their point about the DePT offsets not doing much because they have to handle all tokens more obvious! It also could make this example much clearer where it could cast [t1,t2] as a "system prompt" added after the fact that messes up the learned position embeddings from DePT (and also hightlight how DePT may not play nicely with prompt engineering which ADePT probably would)._`
>
> **A1**: Thank you for your valuable suggestion. Throughout the revised manuscript, we have reframed the weakness of DePT as "position-based embedding offsets.".
>
> ### _**For weakness2**: They state that "embedding for each token should be unique after being offset" as a critique of DePT, but thinking of DePT as position-wise offsets it doesn't seem like a problem, especially given positional embeddings work._
>
> **A2**: Thank you for your valuable suggestion. The idea to update the token embedding matrix first comes from [1], _i.e._, the DePT method. Positional embeddings are added to each layer. Adding offsets only to the input token embedding, treated as input token embedding offsets, should be correct. If DePT is extended to all layers, we think it can be regarded as fine-tuning or adding the position embeddings. I believe that fine-tuning position embeddings in large language models is a worthwhile avenue for research, but it currently seems under-explored.
>
> ### _**For weakness3**: The prose can be tighten up quite a bit. There are lots of parts that repeat themselves multiple times, for example the position-wise implementation of DePT is over-explained multiple times. Similarly, much of the algebraic manipulations of the parameter counts could be omitted._
>
> **A3**: Thank you for your valuable suggestions. In the revised manuscript, we have addressed some problematic paragraphs. Specifically, we have revised the summary in the introduction and the beginning of Section 3.2 and removed the parameter counts in Section 3.3. We are continuing our efforts to make the paper more concise and clear.
>
> [1] Zhengxiang Shi and Aldo Lipani. DePT: Decomposed prompt tuning for parameter-efficient fine-
> tuning. In The Twelfth International Conference on Learning Representations, 2024.

---

> > ### Author Response · Authors · 2024-11-21
> > **Response to Reviewer cKVM (Part 2/4)**
> >
> > ### _**For weakness4**: Many of the increases in performance are rather small, although the simplicity of this method makes that more acceptable._
> >
> > **A4**: Thank you for your valuable question. We have added new experimental results based on CodeGen-350M and Llama3-8B in the revised manuscript. We conduct experiments on MBPP benchmark [1] to test our proposed ADePT on two decoder-only large language models, namely, CodeGen-350M [2] and Llama3-8B [3]. The implementation details are also provided in the revised manuscript. The experimental results are presented as follows, and also in Table 7 in the revised manuscript. We can observe that our proposed ADePT consistently outperforms the vanilla PT and DePT, indicating our proposed method can scale to decoder-only PLMs and larger PLMs. We hope these experiments can make our proposed ADePT more convincing.
> >
> > | Model         | Method    | #Para      | MBPP|
> > |---------------|-----------|------------|----------------------|
> > | **CodeGen-350M** | LoRA      | 1.3M       | 20.32                |
> > |               | PT        | 10.2K      | 16.12                |
> > |               | DePT      | 10.4K      | 16.83                |
> > |               | ADePT(ours) | 10.2K      | 17.86                |
> > | **Llama3-8B** | LoRA      | 9.4M       | 49.08                |
> > |               | PT        | 41.0K      | 18.27                |
> > |               | DePT      | 42.7K      | 42.50                |
> > |               | ADePT (ours) | 41.0k      | 43.22                |
> >
> > ### _**For weakness5**: It is unclear is they use the optimization stabilization of Razdaibiedina et al 2023 they mention in the introduction when using PT._
> >
> > **A5**:  Thank you for your good suggestion. We do not use the optimization stabilization of Razdaibiedina et al 2023 [4]. We believe using this optimization stabilization may help the performance of our proposed ADePT.
> >
> > ### _**For weakness6**: Much of the baseline performance numbers are from other works, opening the possibility of a mis-match in setting. For example, the numbers for SPoT in Table 3 are surprising as they are lower than the Prompt Tuning numbers, but they are taken from different papers._
> >
> > **A6**: Thank you for your valuable question. All baseline methods use the same datasets and backbone model [5,6,7,8]. DePT [5] found that training PT for additional steps typically leads to performance improvements, and we follow this setting. Using this training technique may help improve the performance of SPoT [9]. Reproducing SPoT with this training technique is challenging, and our primary goal is to improve PT rather than using additional multitask learning or transfer learning. In Appendix C of the revised manuscript, we show that we follow the training setting of DePT.
> >
> > [1] Jacob Austin, Augustus Odena, Maxwell Nye, Maarten Bosma, Henryk Michalewski, David Dohan, Ellen Jiang, Carrie Cai, Michael Terry, Quoc Le, et al. Program synthesis with large language models. arXiv preprint arXiv:2108.07732, 2021
> >
> > [2] Erik Nijkamp, Bo Pang, Hiroaki Hayashi, Lifu Tu, Huan Wang, Yingbo Zhou, Silvio Savarese, and Caiming Xiong. Codegen: An open large language model for code with multi-turn program synthesis. In ICLR 2023.
> >
> > [3] Abhimanyu Dubey, Abhinav Jauhri, Abhinav Pandey, Abhishek Kadian, Ahmad Al-Dahle, Aiesha Letman, Akhil Mathur, Alan Schelten, Amy Yang, Angela Fan, et al. The llama 3 herd of models. arXiv preprint arXiv:2407.21783, 2024.
> >
> > [4] Anastasiia Razdaibiedina, Yuning Mao, Madian Khabsa, Mike Lewis, Rui Hou, Jimmy Ba, and Amjad Almahairi. Residual prompt tuning: improving prompt tuning with residual reparameterization. In Anna Rogers, Jordan Boyd-Graber, and Naoaki Okazaki (eds.), In ACL 2023.
> >
> > [5] Zhengxiang Shi and Aldo Lipani. DePT: Decomposed prompt tuning for parameter-efficient fine-tuning. In ICLR 2024.
> >
> > [6] Akari Asai, Mohammadreza Salehi, Matthew Peters, and Hannaneh Hajishirzi. ATTEMPT: Parameter-efficient multi-task tuning via attentional mixtures of soft prompts. In EMNLP 2022.
> >
> > [7] Zhen Wang, Rameswar Panda, Leonid Karlinsky, Rogerio Feris, Huan Sun, and Yoon Kim. Mul-titask prompt tuning enables parameter-efficient transfer learning. In ICLR 2023.
> >
> > [8] Yi-Lin Sung, Jaemin Cho, and Mohit Bansal. LST: Ladder side-tuning for parameter and memory efficient transfer learning. In NeurIPS 2022.
> >
> > [9] Tu Vu, Brian Lester, Noah Constant, Rami Al-Rfou’, and Daniel Cer. SPoT: Better frozen model adaptation through soft prompt transfer. In ACL 2022.

---

> ### Author Response · Authors · 2024-11-21
> **Response to Reviewer cKVM (Part 3/4)**
>
> ### _**For question1**: Did you compare to fine-tuning the prompt and the embedding matrix? The would be a small loss in generality (tokens not seen during training would not get updated) but it would be a much simpler implementation. I would be curious to see how this version performed on CB too, as a possible explanation of ADePT's poor performance could be that the NN offsets don't work well on unattested words._
>
> **A7**: Thanks for your valuable question. Before submitting our manuscript, we did not conduct this experiment. Following your suggestion, based on the T5-base model, we have conducted a preliminary study on simultaneously fine-tuning the prompt matrix and the embedding matrix on the RTE task, as shown below and also shown in Table 11 in Appendix B of the revised manuscript. We use the same length soft prompt, and we search learning rates for prompt matrix from {3e-1, 4e-1, 5e-1} and embedding matrix from {1e-3, 1e-4, 1e-5}. Preliminary experimental results do not seem to support this method, as its performance is significantly worse than our proposed ADePT. Additionally, it requires far more parameters than ADePT. We still believe this is a great idea. This is worth further investigation in the future. The reason might be that fine-tuning with too many parameters can lead to overfitting. Perhaps using LoRA to update the embedding matrix would be beneficial.
>
> For CB test, following the prior works[1,2,3] for few-shot learning, we use MNLI, QQP, SST-2, SQUAD, and ReCoRD as five source tasks to initialize the parameters. These source task datasets are very large, and training the model requires over 30 hours. Due to time and resource constraints, we were unable to complete the testing of this method on CB.
>
> | | #Para | #RTE  |
> | --- | --- | --- |
> | Finetuning prompt and embedding matrix | 24.8M | 76.3 |
> | ADePT | 76.1K | 82.0 |
>
> ### _**For question2**: When measuring the latency of different methods, where the token offsets for ADePT precomputed and folded into the embedding table or was the NN run for each token?_
>
> **A8**: Thanks for your excellent idea. We measured the latency of ADePT by running NN for each token. We believe that precomputing and folding the token offsets into the embedding table can further speed up the inference of ADePT.
>
> ### _**For question3**: Did you try using the LM-adapted model from Lester et al 2021 instead of the span-corrupted T5 models used here? It is also unclear if you used the original T5 models or the T5 1.1 models, IIRC, some of the datasets used were seen during pre-training of the original T5 models._
>
> **A9**: Thanks for your valuable question. We follow the experimental setup of DePT [1]. Table 1 and Table 2 reference experimental results from sources DePT[1], ATTEMPT [2], MPT [3], and LST [4], all of which use the original T5 models rather than the T5 LM-adapted 1.1 version. According to our literature review, the experimental settings of DePT [1] and MPT [3] followed ATTEMPT [2]. ATTEMPT [2] used the original T5 models instead of T5 LM-adapted 1.1 version because it found that T5-LM adapt v1.1 was particularly sensitive and difficult to tune, and thus, ATTEMPT [2] used the original T5 models. In the revised manuscript, we explicitly state in Appendix C that we use the original T5 models rather than T5 LM-adapted 1.1 version
>
> [1] Zhengxiang Shi and Aldo Lipani. DePT: Decomposed prompt tuning for parameter-efficient fine-
> tuning. In The Twelfth International Conference on Learning Representations, 2024.
>
> [2] Akari Asai, Mohammadreza Salehi, Matthew Peters, and Hannaneh Hajishirzi. ATTEMPT: Parameter-efficient multi-task tuning via attentional mixtures of soft prompts. In Proceedings of the 2022 Conference on Empirical Methods in Natural Language Processing, pp. 6655–6672, Abu Dhabi, United Arab Emirates, December 2022. Association for Computational Linguistics.
>
> [3] Zhen Wang, Rameswar Panda, Leonid Karlinsky, Rogerio Feris, Huan Sun, and Yoon Kim. Mul-titask prompt tuning enables parameter-efficient transfer learning. In The Eleventh International
> Conference on Learning Representations, 2023
>
> [4] Yi-Lin Sung, Jaemin Cho, and Mohit Bansal. LST: Ladder side-tuning for parameter and memory efficient transfer learning. In Alice H. Oh, Alekh Agarwal, Danielle Belgrave, and Kyunghyun Cho (eds.), Advances in Neural Information Processing Systems, 2022.

---

> ### Author Response · Authors · 2024-11-21
> **Response to Reviewer cKVM (Part 4/4)**
>
> ### _**For question4**: As the DePT offsets only differ between each position, it seems reasonable that they have low mean and variance. In contrast, ADePT is per-token and content based so it seems reasonable that it would have a much higher variance. I'm not sure mean and variance is the correct metric to gauge how much these offsets are actually doing (or if they are sub-optimized). Something like a norm of the offsets might make more sense? Or it could have been measured directly in an ablation where the soft prompt learned in ADePT/DePT is used without the token offsets._
>
> **A10**: Thank you for your valuable question. In this experiment, we aim to demonstrate that the absolute values of the elements in the token embedding offsets of DePT are very small, while those of ADePT are not. Taking the RTE task as an example, the mean is 0.01 and the variance is 0.06, indicating that the majority of the absolute values of the elements in the token embedding offsets of DePT are likely to be less than 0.1. However, for the input embeddings, the mean is 6.07 and the variance 16.29, indicating the absolute value of elements in input token embeddings is much greater than that of the embedding offsets of DePT with high probability. For the offsets (absolute values) of ADePT, the mean is 8.31 and the variance is 5.45, which is of the same order of magnitude as the input token embeddings. This indicates that ADePT has a broader range of values, meaning there is a significant range where ADePT can achieve what DePT cannot. Additionally, anything that DePT can accomplish, ADePT is also capable of achieving.
>
> We think the norm of the offsets might not make sense as the elements of the offsets. We want to show the elements in the embedding offsets of ADePT have a much larger range of values than that of DePT. The optimal embedding space may lie outside the range of DePT, whereas ADePT may be able to access this embedding space. The sharing of offsets in DePT results in particularly small elements of offsets in DePT. In Section 3.2, we have revised the content to clearly express the ideas.
>
> We show the experimental results of ADePT/DePT when used without the token offsets but only learned soft prompt. We can observe that the token offsets of ADePT play a much more important role than DePT. This experiment is also shown in Table 12 of Appendix B.
> | |With token offsets  |Without token offsets     |
> | --- | --- | --- |
> | DePT| 79.1 | 78.4 |
> | ADePT | 82.0 | 58.3 |

---

### Official Review · Reviewer_znHg · 2024-10-31

**Soundness:** 2
**Presentation:** 3
**Contribution:** 3
**Rating:** 8
**Confidence:** 4

**Summary:**

The paper introduces Adaptive Decomposed Prompt Tuning (ADePT), a novel approach in parameter-efficient fine-tuning (PeFT) that significantly enhances the adaptability of pre-trained large language models (PLMs) to various downstream tasks. ADePT improves Decomposed Prompt Tuning (DePT) by addressing its limitations: DePT's fixed token embedding offsets often underperform due to their inability to dynamically adjust to different model inputs. By integrating a token-shared feed-forward neural network (FFNN), ADePT dynamically adjusts embedding offsets, tailored to each specific input token. This adaptive mechanism allows ADePT to maintain the inference speed advantages of DePT while achieving state-of-the-art (SOTA) performance in adaptation. Extensively tested across 22 natural language processing (NLP) tasks and two PLMs of differing scales, ADePT not only surpasses other PeFT methods like Adapters, LoRA, and standard Prompt Tuning (PT) but also exceeds full model fine-tuning benchmarks in certain scenarios.

**Strengths:**

1. The paper provides a comprehensive overview of Parameter Efficient Finetuning (PeFT), effectively situating ADePT within the broader research landscape and highlighting its contributions.

2. ADePT is intuitive and well-motivated. The arguments and experiments in Section 3.2 convincingly demonstrate the limitations of DePT being a low-rank absolute positional embedding, paving the way for ADePT, which instead uses token-wise MLP for calculating embedding offsets.

3. The experiments conducted with T5-220M are thorough, incorporating all relevant benchmarks and key baseline comparisons. The results are presented with nice detail and clarity.

4. At the 220M scale, ADePT shows remarkable efficacy, particularly in data-scarce scenarios such as RTE and CoLA tasks, where it not only competes but also surpasses full finetuning, showing its superior performance.

**Weaknesses:**

1. The robustness of the experiments with the 3B model does not match the standards set by the 220M scale evaluations. Notably, the selection of fewer benchmark tasks without clear justification, as well as the omission of significant baselines such as Adapters and LoRA, weakens the overall experimental credibility for the 3B model.

2. The performance improvements of ADePT over PT for the T5-3B model is only 0.1 or 0.2 pts for each task in Table 5. This tiny margin on a selected set of tasks may suggest that DePT-style methods cannot scale to larger models.

3. A key appeal of PeFT methods is their cost-efficiency in finetuning; however, the paper lacks comparative analysis of the training costs across different PeFT methods, specifically within the PT family where training speed is more comparable (PT, DePT, and ADePT). Such omission leaves a gap in understanding their real-world benefits.

4. The description of DePT and ADePT in the "Introduction" section lacks clarity. Phrases such as "the token embeddings of DePT violate the uniqueness of token embeddings" are presented without sufficient context, making them hard to understand without referring to the formulas in Section 3.

**Questions:**

1. What are the criteria of selecting the datasets and baselines for 3B-scale models?

---

> ### Author Response · Authors · 2024-11-21
> **Response to Reviewer znHg (Part 1/2)**
>
> Dear Reviewer znHg,
>
> We sincerely thank you for your time and effort in evaluating our manuscript. Your thorough evaluation and insightful comments are highly appreciated. We will address each of your questions point by point and hope to resolve your concerns effectively.
>
> ### _**For weakness 1**: The robustness of the experiments with the 3B model does not match the standards set by the 220M scale evaluations. Notably, the selection of fewer benchmark tasks without clear justification, as well as the omission of significant baselines such as Adapters and LoRA, weakens the overall experimental credibility for the 3B model._
>
> **A1**: Thank you for your valuable suggestion. We first used MNLI and MRPC to test the T5-3B model. The experimental results are as follows:
>
> | | MNLI  | MRPC  |
> | --- | --- | --- |
> | DePT | 89.7  | 90.20 |
> | ADePT | 90.90 | 91.60 |
>
> We found that, in MRPC task, both DePT and ADePT have close performance on T5-3B model compared to T5-base model. However, in MNLI task , we found that the T5-3B model performs much better than the T5-base model. MNLI is a significantly larger dataset than MRPC, as detailed in the Appendix D. Moreover, the T5-3B model needs much more GPU resources than T5-base model. Due to the limitations of computational resources, testing the T5-3B model on all datasets is a difficult task to accomplish. Therefore, we hope to select the most convincing datasets for test. We hope that these datasets have large training and test sets, especially since the diversity of a large test set can provide stronger persuasiveness. Also, we hope the tasks are challenging. These tasks come from three benchmarks: MNLI from the GLUE benchmark (the largest dataset in GLUE benchmark), ReCoRD from the SuperGLUE benchmark (the largest dataset in the SuperGlUE benchmark), and the MRQA 2019 Shared Task, including NLI, Common Sense Reasoning, and Question Answering problems. We use the criteria of more than 70,000 training samples, an accuracy/F1 score of less than 90\% on the T5-base model, and more than 4,000 test samples to select the datasets to test our proposed ADePT on the T5-3B model. I have added some details to Appendix C of the revised manuscript.
>
> We have added the **LoRA** baseline methods in the experiments of the T5-3B model, as presented in **Tables 5 and 6** in the revised manuscript. Since our code implementation is based on the **Peft** library [1], unfortunately, we discovered that the adapter method is not integrated into this library. We plan to add the experimental results of the T5-3B adapters method in the future.
>
> ### _**For weakness 2**: The performance improvements of ADePT over PT for the T5-3B model is only 0.1 or 0.2 pts for each task in Table 5. This tiny margin on a selected set of tasks may suggest that DePT-style methods cannot scale to larger models._
>
> **A2**: Thank you for your valuable feedback. We conduct experiments on MBPP benchmark [2] to test our proposed ADePT on two decoder-only large language models, namely, CodeGen-350M [3] and Llama3-8B [4]. The implementation details are also provided in the revised manuscript. The experimental results are presented as follows, and also in Table 7 in the revised manuscript. We can observe that our proposed ADePT consistently outperforms the vanilla PT and DePT, indicating our proposed method can scale to decoder-only PLMs and larger PLMs. Meanwhile, we can observe that the ADePT method shows a relatively significant improvement on the MBPP benchmark.
>
> | Model         | Method    | #Para      | MBPP|
> |---------------|-----------|------------|----------------------|
> | **CodeGen-350M** | LoRA      | 1.3M       | 20.32                |
> |               | PT        | 10.2K      | 16.12                |
> |               | DePT      | 10.4K      | 16.83                |
> |               | ADePT(ours) | 10.2K      | 17.86                |
> | **Llama3-8B** | LoRA      | 9.4M       | 49.08                |
> |               | PT        | 41.0K      | 18.27                |
> |               | DePT      | 42.7K      | 42.50                |
> |               | ADePT (ours) | 41.0k      | 43.22                |
>
> [1] https://github.com/huggingface/peft
>
> [2] Jacob Austin, Augustus Odena, Maxwell Nye, Maarten Bosma, Henryk Michalewski, David Dohan, Ellen Jiang, Carrie Cai, Michael Terry, Quoc Le, et al. Program synthesis with large language models. arXiv preprint arXiv:2108.07732, 2021
>
>
> [3] Erik Nijkamp, Bo Pang, Hiroaki Hayashi, Lifu Tu, Huan Wang, Yingbo Zhou, Silvio Savarese, and Caiming Xiong. Codegen: An open large language model for code with multi-turn program synthesis. In The Eleventh International Conference on Learning Representations, 2023. URL https://openreview.net/forum?id=iaYcJKpY2B_.
>
> [4] Abhimanyu Dubey, Abhinav Jauhri, Abhinav Pandey, Abhishek Kadian, Ahmad Al-Dahle, Aiesha Letman, Akhil Mathur, Alan Schelten, Amy Yang, Angela Fan, et al. The llama 3 herd of models. arXiv preprint arXiv:2407.21783, 2024.

---

> > ### Author Response · Authors · 2024-11-21
> > **Response to Reviewer znHg (Part 2/2)**
> >
> > ### _**For weakness 3**: A key appeal of PeFT methods is their cost-efficiency in finetuning; however, the paper lacks comparative analysis of the training costs across different PeFT methods, specifically within the PT family where training speed is more comparable (PT, DePT, and ADePT). Such omission leaves a gap in understanding their real-world benefits._
> >
> > **A3**: Thank you for your valuable suggestion. We have added a comparison of training times among LoRA, PT, DePT, and ADePT on Natural Questions and HotpotQA tasks, as shown below. This comparison is also included in Table 10 of the Appendix B of the revised manuscript. Our results indicate that PT, DePT, and ADePT exhibit similar training time. The PT-family method needs longer training time than LoRA due to the longer input sequence. (`''h'' means hours.)
> >
> > | Method    | NQ      | HP|
> > |-----------|------------|----------------------|
> > | LoRA      |   9.73 h    |     9.63 h        |
> > | PT        |    12.60 h   |       12.53 h          |
> > | DePT      |   12.60h    |       12.53 h         |
> > | ADePT(ours) | 12.67 h      |        12.58 h        |
> >
> > ### _**For weakness 4**: The description of DePT and ADePT in the "Introduction" section lacks clarity. Phrases such as "the token embeddings of DePT violate the uniqueness of token embeddings" are presented without sufficient context, making them hard to understand without referring to the formulas in Section 3._
> >
> > **A4**: Thank you for your suggestions on our writing. We have revised the introduction to make it clearer. Please see the details in the revised manuscript.
> >
> > ### _**For question 1**: What are the criteria of selecting the datasets and baselines for 3B-scale models?_
> >
> > **A5**:
> >
> > **For the datasets**:  Thank you for your valuable question. We use the criteria of more than 70,000 training samples, an accuracy/F1 score of less than 90\% on the T5-base model, and more than 4,000 test samples to select the datasets to test our proposed ADePT on the T5-3B model.
> >
> > **For the baselines**: Previous studies of Prompt Tuning have not systematically experimented with the T5-3B model. We chose the most relevant PT methods to our proposed ADePT, and those with the closest number of trainable parameters, for comparison. Following your suggestion, LoRA, a typical PEFT method, should indeed be one of the baseline methods.

---

> ### Comment · Reviewer_znHg · 2024-11-21
> **Thank you for the revised manuscript and response**
>
> Thank you for the revised manuscript and response. I appreciate the new results of ADePT on Llama-3, comparisons against LoRA, and analysis of training time.
>
> However, I still have a few concerns/questions:
>
> 1. ADePT on Llama-3 has only been evaluated on a single task, MBPP, which is quite small (500-ish examples). It would be better if ADePT on Llama-3 is evaluated on more tasks, especially larger ones.
>
> 2. PT-family of methods seem to need more computational resources in the finetuning stage, compared with LoRA. Also, LoRA seems to have better performance on downstream tasks. Is there any appealing reasons why we prefer PT to LoRA?

---

> ### Author Response · Authors · 2024-11-21
>
> Thank you for valuable feedback.
>
> We will address each of your questions point by point and hope to resolve your concerns effectively.
>
> ### **For new question 1**:  ADePT on Llama-3 has only been evaluated on a single task, MBPP, which is quite small (500-ish examples). It would be better if ADePT on Llama-3 is evaluated on more tasks, especially larger ones.
>
> **A6**: Thank you for your valuable question. We will strive to increase experiments with larger datasets on Llama-3.
>
> ### **For new question 2**:  PT-family of methods seem to need more computational resources in the finetuning stage, compared with LoRA. Also, LoRA seems to have better performance on downstream tasks. Is there any appealing reasons why we prefer PT to LoRA?
>
> **A7**:
> Thank you for your valuable question.
>
> We need to clarify one point: the increased training time for PT methods is due to the extended inference time caused by the soft prompt. In contrast, LoRA, which directly merges weights, does not introduce inference latency. LoRA requires far more training parameters than PT methods, which typically results in the need for more GPU resources. Therefore, we cannot simply infer that the training resources required by PT are greater than those required by LoRA based on training time alone.
>
> PT methods have some unique advantages. The number of parameters does not increase with the growth of model layers. In contrast, LoRA requires more parameters as the number of model layers increases. Additionally, LoRA needs to merge model weights, which often limits its applicability to a single downstream task. On the other hand, PT models trained on multiple downstream tasks can be used simultaneously. These points are also mentioned in Section 4.3 and Section 4.4.
>
> The number of parameters required by PT is typically much smaller than that of LoRA. In the experiments already shown,  the ranks of LoRA are 32 for T5-3B or 16 for CodeGen-350M and Llama-3 8B, with the number of parameters being 255 times, 130 times, and 229 times that of the PT methods, respectively. Even with a rank of 1, LoRA requires nearly 10 times more parameters than the PT methods. Regarding this point, we will add experiments with lower-rank LoRA (e.g., rank = 1).

---

> > ### Comment · Reviewer_znHg · 2024-11-21
> >
> > Thank you for your response. I increased my review score increased from 6 -> 8. Looking forward to see more Llama-3 results in the final version.

---

> > > ### Author Response · Authors · 2024-11-23
> > > **Thanks for the reply**
> > >
> > > Dear Reviewer znHg,
> > >
> > > We sincerely thank you for your recognition of this paper and your valuable suggestions.
> > >
> > > In the final version, we will further add more results on Llama-3.
> > >
> > > Thank you for raising your review score. We feel very encouraged.
> > >
> > > Best regards,
> > >
> > > Authors

---

### Official Review · Reviewer_3RMb · 2024-11-03

**Soundness:** 3
**Presentation:** 3
**Contribution:** 3
**Rating:** 6
**Confidence:** 4

**Summary:**

The paper finds that the fixed token embedding offset in DePT limits its generalization capability across different model inputs, leading to suboptimal performance. To address these issues, the authors introduce Adaptive Decomposed Prompt Tuning (ADePT), which consists of a short soft prompt and a shallow token-shared feedforward neural network. ADePT uses the token-shared feedforward neural network to learn embedding offsets for each token. This enables ADePT to achieve superior adaptability without requiring more inference time or additional trainable parameters than standard PT and its variants.

Overall, this paper provides a thorough and clear analysis. It offers a simple yet effective solution to the offset issues present in standard PT. However, I still have the following questions:
1. How robust is this method, and does it possess general applicability?
2. Is this method still effective for long text problems?
3. Is ADePT effective for other large parameter language models, such as the LLaMa series? I hope the authors can provide answers to these questions.

**Strengths:**

This paper introduces the Adaptive Decomposition Prompt Tuning (ADePT) method, which innovatively addresses the generalization limitations caused by fixed token embedding offsets in traditional DePT methods. ADePT achieves excellent adaptability without increasing inference time or requiring additional parameters. The analysis is thorough, and the method is both simple and effective, offering new directions for future research. Additionally, the experiments are conducted rigorously, and the writing is clear and concise.

**Weaknesses:**

I believe this method may lack generality, especially when applied to large language models and long-text tasks. The main reason for questioning this is whether feedforward neural networks possess sufficient semantic understanding capabilities. Additionally, there is room for further optimization in the figure.

**Questions:**

1. How robust is this method, does it have general applicability, and will incorporating AdePT affect the model's generalizability?
2. Is this method still effective for long text problems?
3. Is ADePT effective for other large parameter language models, such as the LLaMa series? I hope the authors can provide answers to these questions.

---

> ### Author Response · Authors · 2024-11-21
> **Response to Reviewer 3RMb**
>
> Dear Reviewer 3RMb:
>
> We sincerely thank you for your time and effort in evaluating our manuscript. Your thorough evaluation and insightful comments are highly appreciated. We will address each of your questions point by point and hope to resolve your concerns effectively.
>
>  ### _**For weakness 1**: I believe this method may lack generality, especially when applied to large language models and long-text tasks. The main reason for questioning this is whether feedforward neural networks possess sufficient semantic understanding capabilities._
>
> **A1**: Thanks for your valuable question. Please see the answers to the questions.
>
> ### _**For weakness 2**:  Additionally, there is room for further optimization in the figure._
>
> **A2**: Thank you for your suggestions regarding the figures. We have improved the design and placement of the figures in the new version of the manuscript, making them more organized and compact.
>
> ### _**For question 1**: How robust is this method, does it have general applicability, and will incorporating AdePT affect the model's generalizability?_
>
> **A3**: Thank you for your valuable question. Inspired by [1], we provide a theoretical analysis towards ADePT, as presented in Section 3.4 of the revised manuscript. In the first transformer layer, the vanilla PT cannot change the relative attention patterns in the first transformer layer across the model input, and can only add a constant bias [1]. Our theoretical analysis shows that in the first transformer layer, our proposed ADePT can change the relative patterns and can also add a bias dependent on the model input. Thus, our proposed ADePT can have more expressive power than the vanilla PT. We believe that our proposed ADePT has a general applicability. For more details, see Section 3.4 of the revised manuscript.
>
> ### _**For question 2**: Is this method still effective for long text problems?_
>
> **A4**: Thank you for your valuable question. We are not familiar with long-text problems. However, I believe that our proposed ADePT can still be effective in the long text fine-tuning tasks because the long text problems still fit our theoretical analysis.
>
> ### _**For question 3**: Is ADePT effective for other large parameter language models, such as the LLaMa series? I hope the authors can provide answers to these questions._
>
> **A5**: Thank you for your valuable question. We conduct experiments on MBPP benchmark [2] to test our proposed ADePT on two decoder-only large language models, namely, CodeGen-350M [3] and Llama3-8B [4]. The implementation details are also provided in the revised manuscript. The experimental results are presented as follows, and also in Table 7 in the revised manuscript. We can observe that our proposed ADePT consistently outperforms the vanilla PT and DePT, indicating our proposed method can scale to decoder-only PLMs and larger PLMs.
>
> | Model         | Method    | #Para      | MBPP|
> |---------------|-----------|------------|----------------------|
> | **CodeGen-350M** | LoRA      | 1.3M       | 20.32                |
> |               | PT        | 10.2K      | 16.12                |
> |               | DePT      | 10.4K      | 16.83                |
> |               | ADePT(ours) | 10.2K      | 17.86                |
> | **Llama3-8B** | LoRA      | 9.4M       | 49.08                |
> |               | PT        | 41.0K      | 18.27                |
> |               | DePT      | 42.7K      | 42.50                |
> |               | ADePT (ours) | 41.0k      | 43.22                |
>
>
>
>
>
> [1] Aleksandar Petrov, Philip Torr, and Adel Bibi. When do prompting and prefix-tuning work? a theory of capabilities and limitations. In The Twelfth International Conference on Learning Representations, 2024.
>
> [2] Jacob Austin, Augustus Odena, Maxwell Nye, Maarten Bosma, Henryk Michalewski, David Dohan, Ellen Jiang, Carrie Cai, Michael Terry, Quoc Le, et al. Program synthesis with large language models. arXiv preprint arXiv:2108.07732, 2021
>
> [3] Erik Nijkamp, Bo Pang, Hiroaki Hayashi, Lifu Tu, Huan Wang, Yingbo Zhou, Silvio Savarese, and Caiming Xiong. Codegen: An open large language model for code with multi-turn program synthesis. In The Eleventh International Conference on Learning Representations, 2023. URL https://openreview.net/forum?id=iaYcJKpY2B_.
>
> [4] Abhimanyu Dubey, Abhinav Jauhri, Abhinav Pandey, Abhishek Kadian, Ahmad Al-Dahle, Aiesha Letman, Akhil Mathur, Alan Schelten, Amy Yang, Angela Fan, et al. The llama 3 herd of models. arXiv preprint arXiv:2407.21783, 2024.

---

> > ### Comment · Reviewer_3RMb · 2024-11-22
> >
> > I read the response and will keep my score unchanged. Thanks.

---

> > > ### Author Response · Authors · 2024-11-23
> > > **Thanks for the reply**
> > >
> > > Dear Reviewer 3RMb,
> > >
> > > We sincerely thank you for your recognition of this paper and your valuable suggestions. We feel very encouraged.
> > >
> > > Best regards,
> > >
> > > Authors

---

### Meta-Review · Area_Chair_mXZe · 2024-12-13

**Metareview:**

The authors propose a method that utilizes a compact MLP with a bottleneck structure, incorporating both down-projection and up-projection, and adds this to the original token embedding. They assess the effectiveness of their approach across various datasets, comparing it against multiple baselines while leveraging several frozen models. Their findings demonstrate that this method consistently outperforms many alternatives in a variety of scenarios.

All reviewers agree that this paper has a positive impact on the field. However, they recommend that the authors revise the paper based on the feedback provided in the reviews.

**Additional Comments On Reviewer Discussion:**

All the reviewers reach a consensus this paper has a positive influence in the field. The authors should revise the paper as the reviewers point out in the reviews.

---

### Decision · Program_Chairs · 2025-01-22

Accept (Poster)